# Dynamic view of the solid-state DNP effect

Deniz Sezer

Institute of Physical and Theoretical Chemistry, Goethe University, 60438 Frankfurt am Main, Germany

**Correspondence:** Deniz Sezer (dzsezer@gmail.com)

**Abstract.** In the solid effect of dynamic nuclear polarization (DNP), the concerted flips of the electronic and nuclear spins, which are needed for polarization transfer, are induced by the microwaves. Commonly, the effect of the microwaves is modeled by a rate process whose rate constant is determined perturbatively. According to quantum mechanics, however, the coherent microwave excitation leads to Rabi nutation, which corresponds to a rotation rather than a rate process. Here we reconcile the coherent effect of the microwaves with the description by rate equations by focusing only on the steady state. We show that the phenomenological rate constants describing the synchronous excitation of the electronic and nuclear spins can be selected such that the description by rate equations yields the same steady state as the exact quantum mechanical treatment. The resulting non-perturbative rates differ from the classical, perturbative ones and apply also at the high microwave powers used in DNP. Our non-perturbative treatment of the solid effect highlights the role of the coherences in the mechanistic steps of polarization transfer, and reveals the importance of the dispersive component of the EPR line. Interestingly, the multiplicative dependence of the DNP enhancement on the dispersive EPR line was intuited in the very first report of the solid effect in liquids [Erb, Motchane and Uebersfeld *Compt. rend.* **246**, 2121 (1958)]. The time-domain description of the solid effect developed here is extendable to liquids, where the dipolar interaction changes randomly in time due to molecular diffusion.

## 1 Introduction

The Boltzmann polarization of electronic spins in a magnetic field is orders of magnitude larger than that of nuclear spins. When the electronic and nuclear spins interact with each other, it becomes possible to transfer the much larger polarization of the former to the latter. Such transfer, known as dynamic nuclear polarization (DNP), can be achieved in several ways, which differ in their mechanistic steps. Two of the DNP mechanisms, namely the Overhauser effect and the solid effect, can be explained by considering a minimal system comprising one electronic spin and one nuclear spin. To explain the other two DNP mechanisms known as the cross effect and thermal mixing, it is necessary to consider one nuclear spin interacting with, respectively, two and many coupled electronic spins (Wenckebach, 2016). The current paper engages only with the former two DNP mechanisms.

Historically, the Overhauser effect was the first to be conceived (Overhauser, 1953) and observed experimentally, initially in metals and subsequently also in liquids (Carver and Slichter, 1953, 1956). A rigorous theoretical understanding of the effect in nonmetals was provided shortly after the first experiments (Abragam, 1955; Solomon, 1955). At the core of this understanding are the Solomon equations, which describe the relaxation processes in a system of two interacting spins (Solomon, 1955).

For our purposes, it is useful to discern two aspects of the theoretical formalism. On the one hand, the evolution of the electronic and nuclear polarizations is described by two coupled differential equations (Solomon, 1955, eq. 14), analogous to the rate equations of chemical kinetics. On the other, the phenomenological rate constants that appear in these rate equations are expressed in terms of the quantum-mechanical probabilities for transition between two distinct energy states (Solomon, 1955, eq. 15). To first order in a perturbative calculation, the amplitude of such transition probabilities per unit time is proportional to the matrix element of the relevant interaction term in the spin Hamiltonian (Solomon, 1955, eq. 3). While the name Solomon equations is mainly used to refer to the first of these aspects (Keeler, 2010), the perturbative calculation of the transition probabilities per unit time is an integral part of the theoretical description. In fact, the idea that interaction terms in the Hamiltonian have corresponding probabilities per unit time to induce transitions (i.e., what we have called the second aspect of the theory), provides the logical justification for the description by rate equations (Abragam, 1955; Solomon, 1955; Webb, 1961).

The solid-state effect (or solid effect) was the second DNP effect that was observed experimentally and explained theoretically (Abragam and Proctor, 1958). In the Overhauser effect, the simultaneous flips of the electronic and nuclear spins, which are needed to couple the electronic and nuclear polarizations, are achieved by thermal relaxation; in the solid effect, these synchronous spin flips are driven coherently by the microwave irradiation. Thus, in the solid effect, the phenomenological rate constants of the rate equations are calculated from the matrix elements of the microwave term in the Hamiltonian. For this term to excite nuclear spin flips, the dipolar interaction between the electronic and nuclear spins should mix the Zeeman energy states, and thus make the zero-quantum (ZQ) and double-quantum (DQ) transitions weakly allowed (Abragam and Proctor, 1958).

Although the Overhauser effect and the solid effect are described using a consistent theoretical formalism (with its two complementary aspects explained above), quantum-mechanically there is a major difference between relaxation and coherent excitation. By their very nature, the rate equations of the polarizations model all evolution as exponential decay/increase towards some steady state. However, according to quantum mechanics, the effect of the microwave field is to rotate the magnetization, leading to the phenomenon known as Rabi nutation. Since rotation and exponential decay/increase are fundamentally different, modeling the effect of the microwaves as a relaxation process should not be possible in general. This raises questions about the fundamental applicability of the first aspect of our theoretical understanding, namely the rate-equation formalism, to the description of the coherently-driven polarization transfer in the solid effect (as opposed to the relaxation-driven transfer in the Overhauser effect). Because the rate equations are justified by the idea that interaction terms induce transitions with a constant probability per unit time, the possibility to model the effect of the microwaves through a perturbative rate constant also becomes questionable. It should be pointed out that these concerns are not new. Indeed, in the case of single spin 1/2, where the quantum dynamics is described exactly by the Bloch equations, Abragam explicitly analyzes how the rate equation with a perturbative rate constant for the microwave (mw) excitation relates to the exact solution, both at short times and at long times (Abragam, 1961, pp. 27-32).

While many modern applications of DNP in the solid state rely on pulsed methods (Can et al., 2015; Quan et al., 2022), here we consider only continuous-wave (cw) excitation, where one is exclusively interested in the steady state of the spin dynamics. As a result, we will be only concerned with how the description of the mw excitation by rate equations relates to its proper

quantum-mechanical description at steady state. To this end, in Sec. 2 we examine the two descriptions for a single spin 1/2 and, following Abragam (1961), confirm that the perturbative rate constant of mw excitation leads to the same steady state as the Bloch equations.

Motivated by this observation, in Sec. 3 we adopt the same perspective to analyze the system composed of one electronic and one nuclear spin 1/2. In this case, starting with the Liouville-von Neumann equation of the density matrix, we first derive proper quantum-mechanical equations of motion for the expectation values of the spin operators that are relevant to the solid effect. Then we show that one can analytically solve for the steady state of the exact quantum dynamics, under the simplifying assumption that the dynamics of the electronic spins is not affected by the hyperfine interaction with the nuclei. Since, at steady

state, all coherences can be expressed in terms of the polarizations, it becomes possible to rewrite the dynamical equations in terms of the polarizations only. Comparing the resulting equations with the rate equations of the polarizations, we select the phenomenological rate constants that appear in the latter, such that the two descriptions have identical steady states.

Stated differently, we abolish the idea of constant transition probabilities per unit time as justification for the rate equations. Instead, we view the rate equations as a convenient mnemonic for encoding the steady state of the exact quantum dynamics,

thus providing a shortcut to the analysis of this steady state. Having decoupled the phenomenological rate constants from the perturbative calculation of the mw-induced transition probabilities, we are free to select them such that the mnemonic yields the correct steady state. We find that the rate constants for the ZQ and DQ transitions selected in this way differ from the corresponding perturbative rate constants that are currently used in the literature (Abragam and Goldman, 1978; Wind et al., 1985; Duijvestijn et al., 1986).

In Sec. 6 we show that our new rate constants reproduce the classical expressions when the Rabi nutation frequency $\omega_1$ is much smaller than the nuclear Larmor frequency $\omega_I$, as required by the perturbative treatment. Our new analytical expressions for driving the forbidden transitions, however, also hold when $\omega_1 > \omega_I$, as could happen at S and X bands, given the high microwave powers currently employed in DNP experiments with resonance structure (Neudert et al., 2016; Denysenkov et al., 2022). These new expressions are the main analytical result of the current paper.

A complete description of the spin dynamics of the four-level system that we analyze here requires only 16 different spin operators, including the identity operator. The dynamics is thus encoded by a $16 \times 16$ propagation matrix in Liouville space, and can be simulated numerically using a spin-dynamics simulation package (Bengs and Levitt, 2018; Yang et al., 2022). Such numerical simulations are currently often employed to explore the efficiency of the solid effect for various experimental parameters. However, even in the relatively simple case of a four-level system, observing a certain effect in the simulations does

not automatically provide understanding about the mechanism of this effect, as demonstrated recently by Quan et al. (2023), who strive to explain the origin of a dispersive DNP component seen both in experiments (Shankar Palani et al., 2023) and in their numerical simulations. Clearly, developing intuition about the spin dynamics that is relevant for a given phenomenon is invaluable.

The general quantum dynamics of a four-level system can be described through 15 coupled differential equations for the

expectation values of the 15 spin operators, excluding the identity. The equations that we derive in Sec. 3.2, together with the Bloch equations from Sec. 2.2, constitute seven such equations. (In fact, we implicitly account for three more operators, thus

covering ten out of the 15 possible ones, as explained in Sec. 4.) When the number of coupled differential equations increases beyond three, gaining an intuitive insight into the dynamics that they describe becomes difficult.

Inspired by the graphical representation of chemical reactions in biochemistry, in Sec. 4 we represent visually the coupled differential equations describing the solid-effect spin dynamics. The resulting "flow diagram" sheds light on the dynamical interconnections between the spin polarizations and the coherences that are active in the solid effect. In Sec. 5 we study the algebraic relationships between the coherences and the polarizations that emerge at steady state. When considered in the context of the dynamical interconnections, these algebraic relationships highlight the importance of the purely electronic coherences in the transfer of polarization, with the out-of-phase (i.e., dispersive) component playing a prominent role. Interestingly, the importance of the dispersive EPR line for the solid effect was intuited already in the first report of the solid effect in liquids (Erb et al., 1958a), as we discuss in Sec. 7.2. Our conclusions are presented in Sec. 7.3.

## 2 Allowed EPR transition

In the rate-equation treatment of the Overhauser and solid effects (Webb, 1961; Barker, 1962), both thermal relaxation and mw excitation are envisioned as randomly flipping spins between pairs of energy levels with certain rates, as depicted in figs. 1a and 1b. The current section aims to illustrate the analytical strategy that we will employ to analyze the solid effect, in the simplest possible case of a single spin 1/2 (fig. 1a). We first present the rate equation of the electronic polarization and obtain its steady state (Sec. 2.1). Then we turn to the Bloch equations and also obtain their steady state (Sec. 2.2). Finally, by requiring that the two descriptions have identical steady states, we identify the rate constant that should be used to describe the effect of the microwaves in the phenomenological rate equation.

### 2.1 Rate equation of the electronic polarization

Let $n_+$ and $n_-$ be the populations of the two energy levels in fig. 1a. Assuming the spins are not destroyed or created, the sum of the two populations is constant in time. Treating the mw excitation as a process that randomly flips the spins with rate constant $v_1$, we have

$$\dot{n}_+|_{\mathrm{mw}} = -\dot{n}_-|_{\mathrm{mw}} = -v_1(n_+ - n_-). \tag{1}$$

(The subscript of the vertical bar indicates that the time derivative accounts only for mw excitation.) Note that $v_1 \geq 0$, since a negative rate constant does not make physical sense.

The electronic spin polarization $P_S = (n_+ - n_-)/(n_+ + n_-)$ is negative at thermal equilibrium, i.e., $P_S^{\mathrm{eq}} < 0$. Differentiating $P_S$ with respect to time and using (1), we find $\dot{P}_S|_{\mathrm{mw}} = -2v_1 P_S$ for the effect of the mw irradiation. The action of thermal relaxation is analogous, after replacing $v_1$ by $w_{1S}$ and taking into consideration that $P_S$ decays towards its thermal equilibrium: $\dot{P}_S|_{\mathrm{th}} = -2w_{1S}(P_S - P_S^{\mathrm{eq}})$. Combining the contributions of mw excitation and thermal relaxation, we get

$$\dot{P}_S = -2v_1 P_S - R_{1S}(P_S - P_S^{\mathrm{eq}}), \tag{2}$$

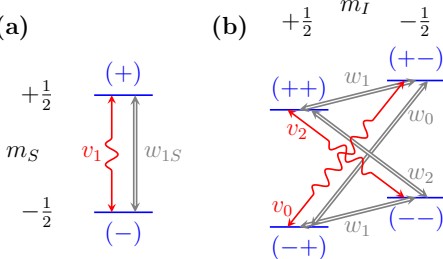

**Figure 1.** Energy levels of (**a**) a single electronic spin $S = 1/2$ and (**b**) one electronic spin and one nuclear spin $I = 1/2$. Microwaves excite single-, zero- and double-quantum transitions (wiggly red arrows) with rate constants $v_1$, $v_0$, and $v_2$, respectively. Thermal relaxation (thick grey arrows) arises from coupling to external degrees of freedom.

where $R_{1S} = 2w_{1S}$. The electronic longitudinal relaxation time is $T_{1S} = 1/R_{1S}$.

In the case of cw irradiation, one is interested in the steady state of the electronic polarization. When the left-hand side of (2) is set equal to zero,

$$P_S^{\text{ss}} = \frac{R_{1S}}{R_{1S} + 2v_1} P_S^{\text{eq}} = p P_S^{\text{eq}}, \tag{3}$$

where the second equality defines the factor $p$. We refer to $p$ as the electronic polarization factor, since it quantifies how close the steady-state polarization is to its Boltzmann value.

The rate equation (2) models the competition between mw pumping and the (longitudinal) relaxation of the polarization. When the two effects balance each other, the polarization is given by the steady-state solution (3). For the rate equation to be a predictive tool, it is necessary to express the phenomenological rate constants $v_1$ and $w_{1S}$ in terms of more fundamental quantities. As discussed in the Introduction, these are identified with the probabilities of transition per unit time between the two energy levels in fig. 1a, which are calculated from time-dependent perturbation theory to first order (Solomon, 1955). In the case of $v_1$, this is basically Fermi's golden rule (Shankar, 1994, Ch. 18), which contains the product of a squared matrix element and a shape-function that accounts for the fact that the energies of the two levels are not infinitely sharp (Abragam, 1961, Sec. II D). For a mw magnetic field in the $x$ direction, the relevant matrix element is $\omega_1 \langle + | S_x | - \rangle$. When the spread of the energy levels is identified with the EPR line shape, which we take to be a Lorentzian, one arrives at

$$v_1(\Omega) = \frac{1}{2} \omega_1^2 \frac{R_{2S}}{R_{2S}^2 + \Omega^2}, \tag{4}$$

where $\Omega = \omega_S - \omega$ is the offset of the mw frequency $\omega$ from the electronic resonance frequency $\omega_S$, and $R_{2S}$ is the electronic $T_2$ relaxation rate.

Formally, this perturbative result is valid only for short times (Cohen-Tannoudji et al., 2019, Ch. XIII). Its validity at long times, including the steady state, thus needs to be explicitly established (Abragam, 1961, pp. 30-32). In the next subsection, we show that (4) is consistent with the steady state of the Bloch equations.

## 2.2 Bloch equations

The effect of the microwaves on the two-level system in fig. 1a is described exactly, and for all times, by the classical Bloch equations. The coherent part of these equations can be derived from the Liouville-von Neumann equation of the density matrix. Specifically, the evolution of the expectation value $q = \langle Q \rangle$ of a general spin operator $Q$, under the action of a spin Hamiltonian $H$ (in units of angular frequency), is

$$\dot{q}|_{\mathrm{coh}} = \mathrm{i}\langle[H,Q]\rangle. \tag{5}$$

We describe the interaction of the electronic spins with the magnetic field using the following Hamiltonian in the rotating frame:

$$H = \Omega S_z + \omega_1 S_x. \tag{6}$$

Here the first term accounts for the Zeeman interaction with the constant magnetic field $B_0$ (along the $z$ axis) and the second for the interaction with the mw field $B_1$ (along $x$).

Using (6) in (5), it is straightforward to obtain the coherent dynamics of $s_z = \langle S_z \rangle$, $s_y = \langle S_y \rangle$ and $s_x = \langle S_x \rangle$. After appending transverse and longitudinal relaxation by hand, one arrives at the familiar Bloch equations

$$\dot{s}_x = -\Omega s_y - R_{2S} s_x$$
$$\dot{s}_y = \Omega s_x - \omega_1 s_z - R_{2S} s_y$$
$$\dot{s}_z = \omega_1 s_y - R_{1S}(s_z - s_z^{\mathrm{eq}}). \tag{7}$$

Since the polarization $P_S$ corresponds to the expectation value of the spin operator $S_z$, the rate equation (2) must be directly comparable to the third equation in (7). However, we see that the effect of the microwaves is modeled differently in the two equations. In the last Bloch equation, the microwaves couple $s_z$ to the transverse component $s_y$. Such coupling is understandably missing in the rate equation, which describes the dynamics of $P_S$ without reference to the transverse components. Clearly, the two descriptions cannot be equivalent in general. Nevertheless, in spite of the fundamentally different ways the two descriptions model the microwaves, there is a regime where the Bloch equations and the rate equation are equivalent, not only approximately but exactly. This is the regime of steady state.

At steady state, the transverse variables $s_{x,y}$ can be eliminated using the first two Bloch equations. From the first equation we find

$$s_x^{\mathrm{ss}} = -\frac{\Omega}{R_{2S}} s_y^{\mathrm{ss}}, \tag{8}$$

where the superscript 'ss' denotes steady state. Substituting this result into the second Bloch equation, we get

$$s_y^{\mathrm{ss}} = -\frac{\omega_1}{R_{2S} + \Omega \frac{1}{R_{2S}} \Omega} s_z^{\mathrm{ss}}. \tag{9}$$

We have thus expressed both transverse components in terms of the longitudinal component as follows:

$$s_{x,y}^{\mathrm{ss}} = \pm(\omega_1 f_{x,y}) s_z^{\mathrm{ss}} \tag{10}$$

(the upper sign corresponds to $x$ and the lower to $y$), where we have defined the auxiliary functions

$$f_y = \frac{1}{R_{2S} + \Omega \frac{1}{R_{2S}}\Omega}, \qquad f_x = \frac{\Omega}{R_{2S}} f_y. \tag{11}$$

Finally, substituting $s_y^{\text{ss}}$ into the third Bloch equation in (7), we arrive at the following differential equation for $s_z$ at steady state:

$$\dot{s}_z^{\text{ss}} = -\omega_1^2 f_y s_z^{\text{ss}} - R_{1S}(s_z^{\text{ss}} - s_z^{\text{eq}}). \tag{12}$$

Although the time derivative on the left-hand side of (12) equals zero, the equation was written in this form to facilitate its comparison with the rate equation (2). Clearly, if the rate constant $v_1$ in (2) is selected such that

$$2v_1 = \omega_1^2 f_y = \omega_1^2 \frac{1}{R_{2S} + \Omega \frac{1}{R_{2S}}\Omega}, \tag{13}$$

then the steady state of $P_S$ will be identical to the steady state of $s_z$. Incidentally, the $v_1$ in (13), which ensures that the two descriptions have the same steady state, is identical to the rate constant obtained from first-order perturbation theory (eq. (4)). This will not be the case for the rate constants of the forbidden transitions, as we show in Sec. 3.

Once the two descriptions are demonstrated to have identical steady states, the analysis of the Bloch equations can be terminated at this point since it will exactly follow the steady-state analysis of the rate equation. In the next section, where we determine the ZQ and DQ transition rates from the steady state of the spin dynamics, we will similarly need to consider only the evolution of $i_z = \langle I_z \rangle$ under the action of the microwaves. The balance between the mw irradiation and the nuclear $T_1$ relaxation will be handled on the level of the rate equation of the nuclear polarization.

For completeness, here we proceed one step further and solve (12) for $s_z^{\text{ss}}$ recalling that the time derivative equals zero. The result is

$$s_z^{\text{ss}} = (R_{1S} f_z) s_z^{\text{eq}}, \tag{14}$$

where we have defined

$$f_z = \frac{1}{R_{1S} + \omega_1^2 f_y}. \tag{15}$$

The functions $f_x$, $f_y$ and $f_z$ introduced in (11) and (15) have units of time, and the factors enclosed in parenthesis in (10) and (14) are dimensionless. (This information is collected in Table 1.) Since (14) is equivalent to (3), it provides an expression for the polarization factor $p = 1 - s$, where $s$ is the familiar saturation factor of the (allowed) electronic transition.

## 3   Forbidden transitions

The excitation of the allowed EPR transition considered above, does not lead to simultaneous flips of the electronic and nuclear spins, and is thus not capable of transferring polarization from the former to the latter. In contrast, the ZQ and DQ transitions involve simultaneous electron-nucleus spin flips (fig. 1b), and drive the solid-state DNP effect. While these, so called, forbidden

**Table 1.** Functions characterizing the steady-state properties of the classical Bloch equations and the Bloch-like equations of the variables $g_n = \langle S_n I_+ \rangle$ $(n = x, y, z)$.

|  | classical Bloch eqs. | Bloch-like eqs. |
| --- | --- | --- |
| unit of time | $f_x, f_y, f_z$ | $F_x, F_y, F_z$ |
| dimensionless | $\omega_1 f_x, \omega_1 f_y, R_{1S} f_z$ | $\omega_1 F_x, \omega_1 F_y, \delta F_z$ |

transitions couple the nuclear and electronic polarizations, their influence on the latter is typically negligible compared to other mechanisms of electronic relaxation. It is therefore justified to write a rate equation for the electronic polarization considering only the allowed EPR transition, as we did in Sec. 2. The effect of the mw-induced ZQ and DQ transitions on the nuclear polarization is described in the current section.

### 3.1 Rate equation of the nuclear polarization

Let $n_{++}$, $n_{+-}$, $n_{-+}$ and $n_{--}$ be the populations of the levels of the four-level system in fig. 1b. While their sum, $n = n_{++} + n_{+-} + n_{-+} + n_{--}$, remains constant in time, the individual populations change due to the ZQ and DQ transitions with rate constants $v_0$ and $v_2$ as follows:

$$\dot{n}_{-+}|_{\text{mw}} = -\dot{n}_{+-}|_{\text{mw}} = -v_0(n_{-+} - n_{+-})$$
$$\dot{n}_{++}|_{\text{mw}} = -\dot{n}_{--}|_{\text{mw}} = -v_2(n_{++} - n_{--}). \tag{16}$$

It is implicitly assumed that $v_0 \geq 0$ and $v_2 \geq 0$, as negative rate constants would not make physical sense.

The polarizations of the nuclear and electronic spins are

$$P_I = [(n_{++} - n_{+-}) + (n_{-+} - n_{--})]/n$$
$$P_S = [(n_{++} - n_{-+}) + (n_{+-} - n_{--})]/n. \tag{17}$$

While, as before, $P_S^{\text{eq}} < 0$, the sign of $P_I$ at thermal equilibrium will depend on the gyromagnetic ratio of the nuclear spin. We will assume protons, hence $\gamma_I > 0$ and $P_I^{\text{eq}} > 0$. Differentiating the definition of $P_I$ in (17) with respect to time, and using (16), we obtain

$$\dot{P}_I|_{\text{mw}} = -v_0(P_I - P_S) - v_2(P_I + P_S)$$
$$= -(v_2 + v_0)P_I - (v_2 - v_0)P_S$$
$$= -v_+ P_I - v_- P_S, \tag{18}$$

which shows that mw excitation of the forbidden transitions couples the evolution of the nuclear polarization to the polarization of the electrons. This coupling is responsible for the solid effect. Because one always encounters either the difference or the sum of $v_0$ and $v_2$, in the third equality of (18) we introduced

$$v_\pm = v_2 \pm v_0. \tag{19}$$

In fact, as we show later, the individual rates $v_0$ and $v_2$ may become negative, and thus meaningless from the rate-equation point of view.

Although in the current paper we are only interested in the rates that describe the effect of the microwaves (i.e., the red arrows in fig. 1), we also discuss thermal relaxation as it is essential for reaching steady state.

Thermal relaxation of the nuclear spins due to their coupling to the electronic spins acts analogously to (18) after replacing the rates $v_{0,2}$ by $w_{0,2}$ and the polarizations by their deviations from thermal equilibrium. Further including nuclear $T_1$ relaxation due to mechanisms other than the coupling to the electrons, we arrive at

$$
\begin{aligned}
\dot{P}_I|_{\text{th}} = {}&-R_{1I}^0(P_I - P_I^{\text{eq}}) - 2w_1(P_I - P_I^{\text{eq}}) \\
&- w_+(P_I - P_I^{\text{eq}}) - w_-(P_S - P_S^{\text{eq}}),
\end{aligned}
\tag{20}
$$

where $R_{1I}^0$ is the nuclear $T_1$ relaxation rate in the absence of the polarizing agent and, analogously to (19),

$$
w_\pm = w_2 \pm w_0.
\tag{21}
$$

The cross-relaxation rate $w_-$ is seen to couple the dynamics of $P_I$ to $P_S$. This coupling leads to the Overhauser effect.

From (20), the total nuclear $T_1$ relaxation rate (i.e., in the presence of the free radical) is identified as $R_{1I} = R_{1I}^0 + 2w_1 + w_+$. Combining the contributions of mw excitation (eq. (18)) and relaxation (eq. (20)), we arrive at the following rate equation for the nuclear polarization:

$$
\begin{aligned}
\dot{P}_I = {}&-R_{1I}(P_I - P_I^{\text{eq}}) - w_-(P_S - P_S^{\text{eq}}) \\
&- v_+ P_I - v_- P_S.
\end{aligned}
\tag{22}
$$

As the rate equations are only used in our analysis to describe the steady state, we solve (22) at steady state and express the nuclear polarization under cw irradiation in terms of the equilibrium polarizations:

$$
P_I^{\text{ss}} = \frac{R_{1I}}{R_{1I} + v_+} P_I^{\text{eq}} + \frac{sw_-}{R_{1I} + v_+} P_S^{\text{eq}} - \frac{pv_-}{R_{1I} + v_+} P_S^{\text{eq}}.
\tag{23}
$$

(We used (3) for the steady-state electronic polarization.)

DNP is generally quantified through the enhancement of the nuclear polarization,

$$
\epsilon = P_I^{\text{ss}}/P_I^{\text{eq}} - 1,
\tag{24}
$$

which is defined such that it equals zero at thermal equilibrium. Taking into account that $P_S^{\text{eq}}/P_I^{\text{eq}} = -|\gamma_S|/\gamma_I$, where $\gamma_S$ and $\gamma_I$ are the gyromagnetic rations of the electronic and nuclear spins, from (23) we obtain

$$
\epsilon = \epsilon_{\text{SE}} + \epsilon_{\text{OE}} + (p_X - 1)
\tag{25}
$$

with

$$
\begin{aligned}
&\epsilon_{\text{SE}} = \frac{pv_-}{R_{1I} + v_+} \frac{|\gamma_S|}{\gamma_I}, \qquad \epsilon_{\text{OE}} = -\frac{sw_-}{R_{1I} + v_+} \frac{|\gamma_S|}{\gamma_I} \\
&p_X = \frac{R_{1I}}{R_{1I} + v_+}.
\end{aligned}
\tag{26}
$$

The first two additive contributions to the DNP enhancement correspond to, respectively, the solid and Overhauser effects. The last one is due to neither of them. Since it does not scale with the ratio of the gyromagnetic ratios, it should be negligible in all cases of practical interest. Note that $p_X$ is similar to the electronic polarization factor $p$ in (3), but with $R_{1S}$ and $2v_1$ replaced by $R_{1I}$ and $v_0 + v_2$.

For the expressions in (26) to have a predictive value, it is necessary to express the rates $v_\pm$ in terms of more fundamental
quantities. This is done using first-order perturbation theory, under the assumption that the dipolar interaction between the electronic and nuclear spins is much smaller than the nuclear splitting (Abragam, 1955). Because the dipolar interaction mixes the Zeeman energy levels depicted in fig. 1b, the ZQ and DQ transitions become weakly allowed. To first order, the mixed states are of the form $(--) + q(-+)$ (Abragam, 1955; Abragam and Proctor, 1958), with mixing parameter

$$q = \frac{1}{4} \frac{D_{\mathrm{dip}}}{\omega_I} \frac{-3\cos\theta\sin\theta\,\mathrm{e}^{\mathrm{i}\phi}}{r^3}. \tag{27}$$

Here, $D_{\mathrm{dip}} = (\mu_0/4\pi)\hbar\gamma_S\gamma_I$ is the dipolar constant, $\gamma_S$ and $\gamma_I$ are the gyromagnetic ratios of the spins, and $(r,\theta,\phi)$ are the spherical polar coordinates of their relative position vector.

The probability amplitude of the microwaves to excite a transition between the mixed energy levels is then proportional to $\omega_1 q$. Combining the probability of excitation with the Lorentzian spread of the electronic energy levels, one arrives at the rate constants (Wind et al., 1985)

$$v_{0,2}(\Omega) = 4(q^*q)\,v_1(\Omega \pm \omega_I), \tag{28}$$

where $v_1$ is the rate of the allowed (single-quantum) EPR transition (eq. (4)). In essence, the rates of the ZQ and DQ transitions are obtained by shifting the rate of the allowed transition along the frequency axis by $\pm\omega_I$, and reducing its magnitude through multiplication by $4|q|^2$.

We observe that in this approach the rates of the forbidden transitions acquire a factor of $\omega_I^{-2}$ from $|q|^2$, and a factor of $\omega_1^2$
from the mw excitation (eq. (4)), without any room for non-trivial cross-talk between these two frequencies. Such cross-talk is also not provided by the Lorentzian dependence on $\Omega$. Like (28), the rate constants that we will obtain in the next subsection will also contain $\omega_1^2$ and $D_{\mathrm{dip}}^2$ as multiplicative factors. However, their offset dependence will couple $\omega_1$ and $\omega_I$ in a non-trivial way, which reduces to the classical expression when $\omega_1 \ll \omega_I$ but predicts qualitatively different dependence when $\omega_1$ is similar to or larger than $\omega_I$ (Sec. 6.1).

## 3.2 Generalized Bloch equations for the solid effect

In this section, we obtain alternative expressions for the forbidden-transition rates $v_\pm$ from the steady state of the exact quantum dynamics. We start by deriving equations of motion for the expectation values of the operators relevant to the solid effect. To use (5), we need to first specify the Hamiltonian guiding the dynamics.

We will consider the minimal solid-effect spin Hamiltonian (Wenckebach, 2016)

$$H = \Omega S_z + \omega_1 S_x - \omega_I I_z + \frac{1}{2}(A_1^* S_z I_+ + A_1 S_z I_-), \tag{29}$$

which is in the rotating frame for the electronic spin and in the laboratory frame for the nuclear spin. The first two terms are the same as in the Hamiltonian (6). The third term describes the nuclear Zeeman interaction. The sign of $\omega_I$ is negative since we assumed a nuclear spin with positive gyromagnetic ratio. The last two terms in (29) account for the dipolar interaction between the electronic and nuclear spins. We have truncated this interaction by dropping all non-secular terms containing $S_x$ and $S_y$. Similar to the assumption behind the derivation of the mixing factor (eq. (27)), we take the dipolar interaction to be small compared to the nuclear Zeeman splitting and drop the secular term proportional to $S_z I_z$. The remaining, pseudosecular terms scale with the dipolar coupling (Wenckebach, 2016)

$$A_1 = D_{\mathrm{dip}} \frac{-3\cos\theta\sin\theta}{r^3} e^{i\phi} \tag{30}$$

where $D_{\mathrm{dip}}/2\pi \approx 79.066\,\mathrm{kHz\,nm^3}$ for protons. The subscript of $A_1$ indicates that its angular dependence is identical to the second-degree spherical harmonic of order $m = 1$.

We start our derivation of equations of motion with $i_z = \langle I_z \rangle$, as it corresponds to the nuclear polarization. There is no contribution from the first three terms in the Hamiltonian (29) as $I_z$ commutes with all of them (eq. (5)). From the commutator with the dipolar terms we obtain

$$\dot{i}_z|_{\mathrm{coh}} = i\frac{1}{2}(A_1 g_z^* - A_1^* g_z) = -\mathrm{Re}\{iA_1^* g_z\}, \tag{31}$$

where

$$g_n = \langle S_n I_+ \rangle \qquad (n = x, y, z). \tag{32}$$

Proceeding in the same way, we first find

$$\dot{g}_z|_{\mathrm{coh}} = -i\omega_I g_z + \omega_1 g_y - i(A_1/4)i_z \tag{33}$$

and then

$$\dot{g}_y|_{\mathrm{coh}} = \Omega g_x - i\omega_I g_y - \omega_1 g_z + (A_1/4)s_x$$
$$\dot{g}_x|_{\mathrm{coh}} = -i\omega_I g_x - \Omega g_y - (A_1/4)s_y. \tag{34}$$

The chain of dynamical equations can be terminated at this stage, as $s_{x,y}$ obey the classical Bloch equations discussed above. (The dynamics of the electronic spin was taken to be independent of its dipolar coupling with the nuclei.)

In addition to the coherent evolution considered so far, $g_z = \langle S_z I_+ \rangle$ and $g_{x,y} = \langle S_{x,y} I_+ \rangle$ are expected to decay with rates $R_{1S} + R_{2I}$ and $R_{2S} + R_{2I}$, respectively. Neglecting $R_{2I}$ compared to $R_{1S}$ and $R_{2S}$, we arrive at the following system of coupled differential equations:

$$\dot{g}_x = -(R_{2S} + i\omega_I)g_x - \Omega g_y - (A_1/4)s_y$$
$$\dot{g}_y = \Omega g_x - (R_{2S} + i\omega_I)g_y - \omega_1 g_z + (A_1/4)s_x$$
$$\dot{g}_z = -(R_{1S} + i\omega_I)g_z + \omega_1 g_y - i(A_1/4)i_z. \tag{35}$$

Equations (31) and (35), supplemented by the Bloch equations (7), constitute the generalization of the Bloch equations to the four-level system in fig. 1b as relevant to the solid effect. If desired, one can also supplement (31) with nuclear $T_1$ relaxation. However, because our aim is to identify the rates $v_\pm$, this is not necessary. In any case, the balance between thermal relaxation and mw excitation at steady state was already analyzed using the rate-equation formalism (Sec. 3.1).

Analogously to our treatment of the Bloch equations (Sec. 2.2), we will now use the condition of steady state to eliminate all variables except the polarizations $i_z$ and $s_z$. From the steady state of the first equation in (35) we get

$$g_x^{\text{ss}} = -\frac{\Omega}{R_{2S} + \text{i}\omega_I} g_y^{\text{ss}} - \frac{A_1/4}{R_{2S} + \text{i}\omega_I} s_y^{\text{ss}}. \tag{36}$$

Substituting into the second equation of (35) we find

$$g_y^{\text{ss}} = -\omega_1 F_y g_z^{\text{ss}} + (A_1/4)(F_y s_x^{\text{ss}} - F_x s_y^{\text{ss}}), \tag{37}$$

where we introduced the complex-valued functions

$$F_y = \frac{1}{R_{2S} + \text{i}\omega_I + \Omega \frac{1}{R_{2S}+\text{i}\omega_I}\Omega}$$

$$F_x = \frac{\Omega}{R_{2S} + \text{i}\omega_I} F_y, \tag{38}$$

which generalize the functions (11) of the classical Bloch equations by supplementing their relaxation rates with an imaginary part. Like their real analogs, $F_{x,y}$ have units of time (Table 1).

Substituting $g_y^{\text{ss}}$ into the last equation of (35) and solving for $g_z$ at steady, we find

$$g_z^{\text{ss}} = -\text{i}(A_1/4)F_z i_z^{\text{ss}} + (A_1/4)F_z(\omega_1 F_y s_x^{\text{ss}} - \omega_1 F_x s_y^{\text{ss}}), \tag{39}$$

where the function

$$F_z = \frac{1}{R_{1S} + \text{i}\omega_I + \omega_1^2 F_y} \tag{40}$$

generalizes (15) of the classical Bloch equations. Finally, we substitute $g_z^{\text{ss}}$ into the equation of $i_z$ (eq. (31)). Factoring out the dipolar coupling as

$$\delta^2 = (A_1^* A_1)/4, \tag{41}$$

at steady state (31) becomes

$$\dot{i}_z^{\text{ss}}|_{\text{coh}} = -\delta^2 \text{Re}\{F_z\} i_z^{\text{ss}} - \delta^2 \text{Re}\{\text{i}F_z(\omega_1 F_y)\} s_x^{\text{ss}} - \delta^2 \text{Re}\{\text{i}F_z(-\omega_1 F_x)\} s_y^{\text{ss}}. \tag{42}$$

We have thus managed to eliminate the three electron-nucleus coherences $g_n$.

To further eliminate the electronic coherences from (42), we recall that at steady state the transverse components $s_{x,y}$ are algebraically related to $s_z$ (eq. (10)). Hence,

$$\dot{i}_z^{\text{ss}}|_{\text{coh}} = -\delta^2 \text{Re}\{F_z\} i_z^{\text{ss}} - \delta^2 \omega_1^2 \text{Re}\{\text{i}F_z(F_y f_x + F_x f_y)\} s_z^{\text{ss}}. \tag{43}$$

As the right-hand side of (43) contains only $i_z$ and $s_z$, it can be directly compared with the rate equation (18), which accounts for the contribution of the microwaves to the time derivative of $P_I$. The comparison allows us to identify the two phenomenological rate constants of the forbidden transitions as

$$v_+ = \delta^2 \text{Re}\{F_z\}, \quad v_- = \delta^2 \omega_1^2 \text{Re}\{iF_z(F_y\, f_x + F_x\, f_y)\}. \tag{44}$$

When used with these two rate constants, the rate equation of $P_I$ is guaranteed to have the correct steady state.

The above non-perturbative derivation of the rate constants $v_\pm$ is the main analytical contribution of the current paper. In Sec. 6, we will explore the predictions of these expressions, as well as their relationship with the classical perturbative rates (eq. (28)). Before that, in the next section, we revisit the equations of motion (31), (35), and the Bloch equations (7), which constitute a system of seven coupled differential equations. The steady state of this system of equations is examined in Sec. 5.

## 4 Making sense of the spin dynamics

The Bloch equations (eq. (7)) are coupled differential equations describing the time evolution of three dynamical variables. When the number of coupled equations is small, it is possible to form a mental picture of the dynamical interconnections between the variables by examining the written equations. In the case of more than three variables, however, gaining insight into the dynamics by simply looking at the written equations becomes harder.

The need to make sense of several coupled differential equations also arises in the context of chemical reaction kinetics, where the concentrations of the reactants change in time. When the number of chemical species is small, it is sufficient to write down the kinetic equations for the concentrations. However, when one deals with the reactions of even a relatively simple metabolic pathway, like glycolysis or the citric acid cycle, the rate equations are almost never written down explicitly. Instead, they are represented in a visual way by drawing arrows between the names of the chemical species that are interconverted by the reactions.

Following the same logic, we represent the dynamical variables $s_x$, $s_y$ and $s_z$ of the classical Bloch equations (7) as nodes, and the various interactions that couple their dynamics as arrows (fig. 2a). The time derivative of each variable is calculated by summing the contributions of all arrows that point *into* its node, where the contribution of an arrow is obtained by multiplying the weight of the arrow by the variable from which it *originates*. Differently from the representation of chemical reactions, here an arrow does not deplete the node at its origin but only contributes to the node at its pointed end. In addition, as our arrows do not have the physical interpretation of reaction rate constants, their weights may also be negative.

The two orange arrows in fig. 2a, which flow into the node of $s_x$, correspond to the two terms on the right-hand side of the first Bloch equation in (7). The arrow with weight $-\Omega$ originates from $s_y$, and thus contributes $-\Omega s_y$ to the time derivative of $s_x$. The other orange arrow originates from $s_x$ and accounts for the decay of this variable with the rate constant $R_{2S}$ of the transverse relaxation. We refer to such arrows that leave a node and enter the same node as self-arrows. To prevent positive feedback, and thus ensure dynamical stability, the total contribution of self-arrows (in case several such arrows point into a node) should be positive. We will generally write the weight of a self-arrow with an explicit negative sign, which we place inside the loop formed by the arrow.

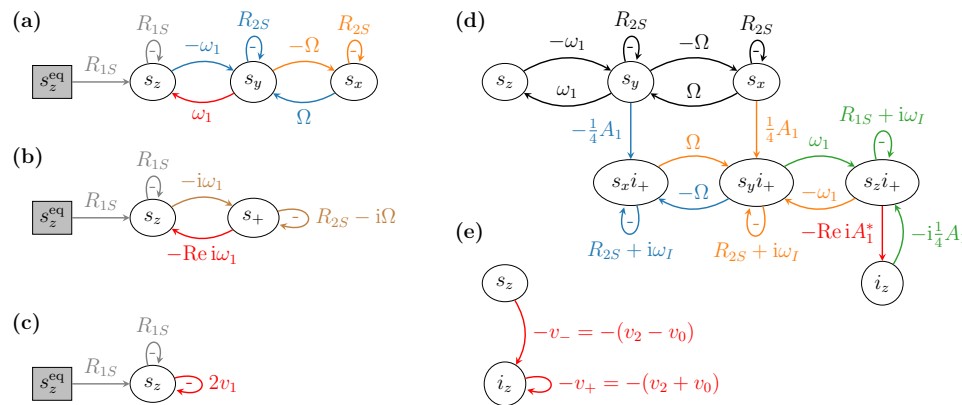

**Figure 2.** (**a**) Real-valued and (**b**) complex-valued classical Bloch equations and (**c**) corresponding dynamics according to the rate equation of the electronic polarization. (**d**) Spin dynamics of relevance to the solid effect and (**e**) corresponding dynamics implied by the rate equation of the nuclear polarization.

Similarly, the three blue arrows in fig. 2a, which flow into the node of $s_y$, correspond to the three terms on the right-hand side of the second Bloch equation in (7). The remaining three arrows, which flow into $s_z$, correspond to the right-hand side of the last Bloch equation. Rather then using the same color for these three arrows, we have indicated the contribution of mw irradiation with red and the contribution of relaxation with gray, in line with the colors used in fig. 1a. In any case, the colors of the arrows do not play a role in the correspondence between the differential equation and its visual representation. Because the equilibrium value $s_z^{\mathrm{eq}}$ is a constant parameter in the Bloch equations, there are no arrows flowing into its node. A node is shaded gray when the corresponding variable remains constant in time.

While fig. 2a contains exactly the same information as the Bloch equations (7), all dynamical interconnections between the variables are now visually accessible. For example, the loop formed by the two arrows with weights $-\omega_1$ and $\omega_1$ between the variables $s_y$ and $s_z$, corresponds to rotation in the $y$-$z$ plane with angular velocity equal to $\omega_1$. In other words, this loop is a visual manifestation of the Rabi nutation driven by the microwaves. There is a similar loop between the variables $s_x$ and $s_y$, which corresponds to rotation with angular velocity $\Omega$ in the $x$-$y$ plane. This is the Larmor precession, as seen in the rotating frame. Since all other arrows correspond to relaxation, the diagram in fig. 2a confirms in a visual way that the coherent part of the Bloch equations consists of two rotations.

At this point we mention that instead of working with the real-valued Bloch equations (7), one could form the dynamical variable $s_+ = s_x + \mathrm{i}s_y$ and work with the complex-valued Bloch equations

$$\dot{s}_+ = -(R_{2S} - \mathrm{i}\Omega)s_+ - \mathrm{i}\omega_1 s_z$$
$$\dot{s}_z = -R_{1S}(s_z - s_z^{\mathrm{eq}}) - \mathrm{Re}\{\mathrm{i}\omega_1 s_+\}. \tag{45}$$

These two differential equations are depicted in fig. 2b. Notably, the rotation in the $x$-$y$ plane with angular velocity $\Omega$ (i.e., the Larmor precession) has now become the imaginary part of the self-arrow of $s_+$ whose real part is the $T_2$ relaxation rate.

In (45) we arbitrarily retained $s_+$ and dropped $s_-$, thus reducing the number of variables in the diagrammatic representation from three to two. (The analogous reduction will be more substantial in the case of the coupled electron-nucleus system.) Note, however, that the contribution of $s_-$ is recovered when the real part of $is_+$ is evaluated to calculate the time derivative of $s_z$ in the second line of (45).

In fig. 2c we have represented the dynamics of $s_z$ which is implied by the rate equation of the electronic polarization (eq. (2)). The visual comparison of this dynamics with the Bloch equations above it makes clear that the rate $v_1$ of the allowed EPR transition is supposed to account in some effective way for the coupling between $s_z$ and $s_y$ (due to $\omega_1$), and for the dynamics of the transverse components (due to $\Omega$ and $R_{2S}$). Indeed, the rate constant $v_1$ in (13) is a function of $\omega_1$, $\Omega$ and $R_{2S}$.

In fig. 2d we show the system of seven coupled differential equations that play a role in the solid effect (eqs. (31), (35) and the Bloch equations). For clarity, the nodes of $g_n$ ($n = x, y, z$) are labeled as $s_n i_+$ in the figure. Black arrows correspond to the classical Bloch equations. Blue, orange and green arrows, which flow into the nodes $g_x$, $g_y$ and $g_z$, respectively, correspond to the right-hand sides of the three equations in (35). The red arrow flowing into the node of $i_z$ corresponds to the right-hand side of (31). Note that the weight of the red arrow involves taking a real part, just like in the complex-valued Bloch equations. Thus, although we only show the dynamics of the coherences $S_n I_+$, at this point the effect of the coherences $S_n I_-$ is also included. In other words, if we did not take the real part, we would need to represent ten coupled differential equations, rather than seven.

The graphical representation of the spin dynamics in fig. 2d lays bare the overall topology of the dynamical connections between the seven variables. For example, note that the Bloch-equations pattern connecting the top three nodes (black arrows) is recapitulated between the nodes of the coherences $g_n$ below them. Indeed, between the electron-nucleus coherences one recognizes the loops that correspond to Rabi nutation and Larmor precession. Due to the involvement of the nuclear spin operator $I_+$, this second set of Bloch equations is "shifted" by the nuclear Larmor frequency, as evidenced by the imaginary part of the self-arrows of $g_n$. The link between the electronic Bloch equations and these new Bloch equations that describe the dynamics of the $S$-$I$ coherences is established by the dipolar coupling ($A_1$), which connects the two sets of Bloch equations such that the $y$ variable of one of them feeds into the $x$ variable of the other, and vice versa. The same dipolar interaction also connects $g_z$ to the nuclear polarization through the red arrow in fig. 2d. Although the coherences $S_n I_-$ are not explicitly modeled, their contribution is recovered when we feed a real value into the time derivative of $i_z$, as discussed above. At this stage, Bloch-like equations shifted by $+\omega_I$ (shown) and by $-\omega_I$ (not shown) contribute symmetrically to the nuclear polarization.

All interactions in the Hamiltonian (29) lead to rotations, which are manifested as loops between two variables formed by arrows with opposite weights. Although such loops are also formed between the variables $s_y$ and $g_x$, and between $s_x$ and $g_y$, we have not shown the arrows that originate at $g_x$ and $g_y$ and flow into, respectively, $s_y$ and $s_x$. These arrows, which would complete the loops of the dipolar interaction, are dropped because their contribution to the electronic dynamics is neglected.

For comparison, in fig. 2e we recall the description of the same spin dynamics according to the rate-equation formalism (eq. (18)). Clearly, the two rates $v_\pm$ should summarize in some faithful way the complexity of the proper, quantum-mechanical dynamics in fig. 2d. In particular, the rate $v_-$ should account for the pathways from $s_z$ to $i_z$, and the rate $v_+$ for the pathways from $i_z$ into the coherences $g_n$ and back to $i_z$.

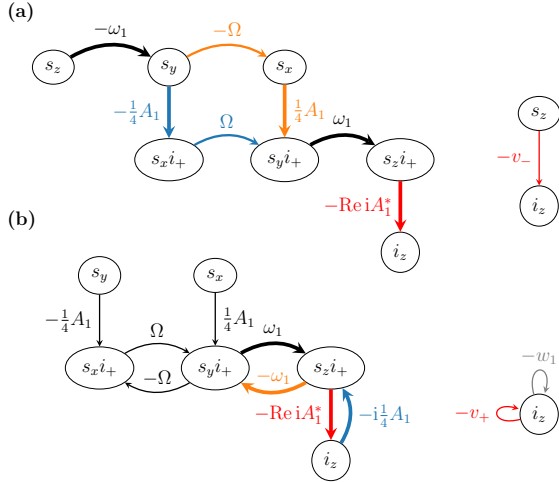

**Figure 3.** Pathways (a) from $s_z$ to $i_z$ contributing to the rate $v_-$ and (b) from $i_z$ to $i_z$ contributing to the rate $v_+$.

By examining the pathways from $s_z$ to $i_z$ we gain visual understanding of the mechanism of dynamical coupling between the electronic and nuclear polarizations in the solid effect. The two possible paths for reaching $i_z$ from $s_z$ following the "flow" of the arrows are shown in fig. 3a. Both paths consists of four steps, not counting the mixing of the transverse components by $\pm\Omega$. First, the mw excitation ($\omega_1$) generates the transverse components $s_{y,x}$ from $s_z$. This step is described by the classical Bloch equations. Then, the dipolar coupling ($A_1$) generates the coherences $g_{x,y}$ from $s_{y,x}$. These are then converted to $g_z$ by the mw excitation, and finally the dipolar interaction transforms $g_z$ to $i_z$. Observe that the weights $\omega_1$ and $A_1$ appear twice along each path, thus both paths scale as $\omega_1^2|A_1|^2$. Since these paths contribute to the rate constant $v_-$, it should also scale with the mw power and the square of the dipolar interaction, which is in agreement with the perturbative rates in (28).

In addition to the weights considered above, both paths in fig. 3a also traverse arrows with weights $\pm\Omega$. Thus, on resonance (i.e., $\Omega = 0$), the possibility of polarization transfer is severed. This observation does not appear to be particularly useful as the forbidden transitions are driven at $\Omega \approx \pm\omega_I$ anyway. However, since going along an arrow with weight $\pm\Omega$ amounts to multiplication by $\Omega$, we realize that crossing from the left side of the dynamical network to the right side involves change of parity in $\Omega$. In other words, because $s_z$ is an even function of the frequency offset, its effect on $i_z$ must be odd in $\Omega$. This is the reason for the anti-symmetric field profile of the solid effect (in contrast to the symmetric profile of the Overhauser effect). The diagram makes clear that the solid effect is odd in $\Omega$ for the same reason that $s_x$ is odd. This point is further examined in Sec. 5.

In fig. 3b we have highlighted the arrows that contribute to the self-loop of $i_z$ with weight $v_+$. Again there are two different possible paths: one consists of two steps and the other of four. The shorter path from $i_z$ to $g_z$ (blue arrow), and back to $i_z$ (red arrow), relies only on the dipolar interaction between the electronic and nuclear spins, and must be active even in the absence of mw excitation. The longer path additionally goes from $g_z$ to $g_{x,y}$ (the latter are mixed by $\Omega$) and back, and contributes only under mw irradiation. Considering that thermal relaxation and mw excitation are treated separately, we realize that the short

loop in fact contributes to the nuclear $T_1$ relaxation (more precisely to the rate $w_1$ in fig. 1b), hence its contribution should be removed when calculating the rate $v_+$.

On the basis of this observation, we now modify the analytical expression for $v_+$ that we gave in (44). Since the nuclear $T_1$ is typically measured with the microwaves switched off, we identify the $\omega_1$-independent part of $F_z$ (eq. (40)), namely

$$F_z(\omega_1 = 0) = (R_{1S} + \mathrm{i}\omega_I)^{-1}, \tag{46}$$

as contributing to relaxation. The corrected form of the first equality in (44) is thus

$$v_+ = \delta^2 \mathrm{Re}\{F_z - (R_{1S} + \mathrm{i}\omega_I)^{-1}\}. \tag{47}$$

Having a visual representation of the spin dynamics was thus helpful to identify an aspect that would be harder to identify on the level of the written equations.

## 5  Analyzing the steady state

The diagrammatic representations of the previous section showed that the quantum-mechanical dynamics consists of several simultaneous rotations that mix the expectation values of the various spin operators. In spite of the complicated time evolution that such interconnected rotations may generally lead to, relatively simple algebraic relationships between the variables emerged at steady state (Secs. 2.2 and 3.2).

The steady-state relationships of the Bloch equations, which were given in (10) and (14), are depicted diagrammatically in fig. 4a. Because we deal with algebraic (as opposed to differential) equations, the inflowing arrows now contribute directly to the value of the variable inside the node, and not to its time derivative. To make this distinction visually clear, we use a rectangular node when the variable itself is obtained by adding the contributions of all inflowing arrows. In addition, we use dashed arrows to signal that the mathematical relationships hold only at steady state. In contrast, the solid arrows of the previous section represented fundamental, causal relationships between the variables governing their dynamics at all times.

It is convenient to think of the steady-state Bloch equations as a system that takes the Boltzmann polarization $s_z^{\mathrm{eq}}$ as an input and produces the outputs $s_{x,y}^{\mathrm{ss}}$, as suggested graphically in fig. 4a. Each dashed arrow can thus be viewed as a transfer function that multiplies the variable at its input to produce the variable at its output. The weights of the arrows in fig. 4a are dimensionless (Table 1).

Equation (43), from which we identified the rates $v_\pm$, is depicted in fig. 4b and, equivalently, in fig. 4c. The three colored arrows in fig. 4b correspond to the three terms on the right-hand side of (42), before the transverse components $s_{x,y}^{\mathrm{ss}}$ were replaced by $s_z^{\mathrm{ss}}$. Specifically,

$$T_i = \mathrm{Re}\{F_z\}, \quad T_x' = \mathrm{Re}\{\mathrm{i}F_z(\omega_1 F_y)\}, \quad T_y' = \mathrm{Re}\{\mathrm{i}F_z(-\omega_1 F_x)\}. \tag{48}$$

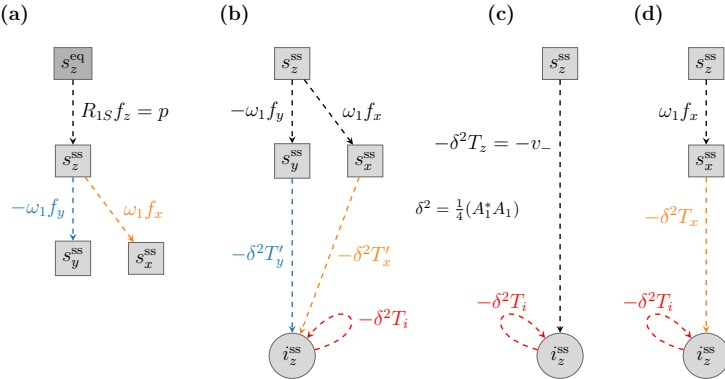

**Figure 4.** Algebraic relations between the dynamical variables at steady state. (**a**) Transfer functions of the Bloch equations. (**b, c, d**) Transfer functions describing the steady-state relationship between the time derivative of $i_z$ (output) and different choices of the electronic input.

The cumulative transfer function from $s_z$ to $i_z$ (fig. 4c) is obtained by adding the contributions of the two parallel paths in fig. 4b. The sum

$$
\begin{aligned}
T_z &= \omega_1 f_x T'_x - \omega_1 f_y T'_y \\
&= \omega_1^2 \operatorname{Re}\{iF_z(F_y f_x + F_x f_y)\} = v_- / \delta^2
\end{aligned}
\tag{49}
$$

was already evaluated in (43).

## 5.1 Bloch equations

To examine the steady-state properties of the Bloch equations, in fig. 5 we plot the ratios $s_z^{\mathrm{ss}}/s_z^{\mathrm{eq}} = p(\Omega, \omega_1)$ (first row) and $s_{x,y}^{\mathrm{ss}}/s_z^{\mathrm{ss}} = \pm\omega_1 f_{x,y}(\Omega)$ (second row) against the offset frequency $\Omega$ for four different values of $B_1$. A free radical with $g = 2$ was assumed when converting $B_1$ to $\omega_1$, so that $B_1 = 6\,\mathrm{G}$ corresponds to $\omega_1/2\pi = 16.8\,\mathrm{MHz}$. This maximum value of $B_1$ is intended to reflect the actual mw field of modern-day DNP spectrometers at X band (Neudert et al., 2016) and at J band (Kuzhelev et al., 2022). The electronic relaxation times used in the plots were $T_{2S} = 60\,\mathrm{ns}$ and $T_{1S} = 9\,T_{2S} = 540\,\mathrm{ns}$.

From the first row of fig. 5 we see that the electronic saturation is most efficient on resonance ($\Omega = 0$) and quickly becomes inefficient at larger offsets. With increasing mw power (different columns) the deviation of $s_z^{\mathrm{ss}}$ from equilibrium spreads to larger offsets. As our main interest is in the solid effect, we have indicated with dashed vertical lines the offsets $\Omega$ that correspond to proton Larmor frequencies at the X (9.2 GHz/14 MHz), Q (30 GHz/45 MHz) and W (92 GHz/140 MHz) mw bands. Considering that DNP is performed at high mw powers, let us examine the saturation at $B_1 = 6\,\mathrm{G}$ (fig. 5, upper right plot).

Looking at $\Omega = \omega_I$ at X band, we see that the allowed EPR transition is almost completely saturated. Because the efficiency of the solid effect scales with $p$ (eq. (26)) any gain from efficiently driving the forbidden transitions will be squashed down

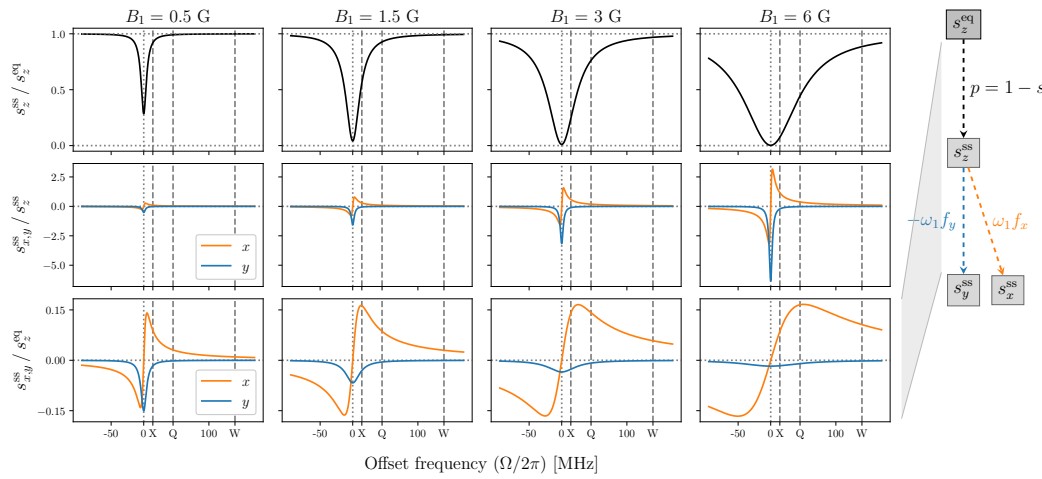

**Figure 5.** Transfer functions characterizing the steady state of the classical Bloch equations. The conversion of $B_1$ to $\omega_1$ was for free radical with $g = 2$, hence $B_1 = 6\,\text{G}$ corresponds to $\omega_1/2\pi = 16.8\,\text{MHz}$. In all plots $T_2 = 60\,\text{ns}$ and $T_1 = 9\,T_2$. The positions of the nuclear Larmor frequencies at X (14 MHz), Q (45 MHz) and W (140 MHz) bands are indicated with vertical dashed lines.

dramatically, thus substantially reducing the ultimate enhancement of the NMR signal. This observation implies that at X band the best solid-effect enhancement may occur at less than maximum mw power, as we demonstrate numerically in Sec. 6.2.

The second row of fig. 5 shows the offset dependence of the transfer functions connecting the longitudinal component $s_z^{\text{ss}}$
to the transverse components $s_{x,y}^{\text{ss}}$. The observed increase in magnitude from left to right reflects the multiplication by $\omega_1$ of the functions $f_{x,y}$ which are independent of $\omega_1$ (eq. (11)). Being the real ($f_y$) and imaginary ($f_x$) components of a complex-valued Lorentzian with width $R_{2S}$ and center frequency $\Omega = 0$, these functions correspond to the absorptive and dispersive components of a homogeneous EPR line. The absorptive component (blue line) is largest at $\Omega = 0$, while the two extrema of the dispersive component (orange line) are located at $\Omega = \pm R_{2S}$. At offsets much larger than the locations of these extrema
(i.e., $\Omega \gg R_{2S}$), the absorptive component drops as $1/\Omega^2$ while the dispersive component drops as $1/\Omega$.

The third row of fig. 5 shows the net transfer functions relating the input of the Bloch equations, $s_z^{\text{eq}}$, to their ultimate outputs, $s_{x,y}^{\text{ss}}$. These transfer functions are obtained by multiplying the solid black lines in the first row with the lines in the second row. In essence, what we see are the absorptive and dispersive components of a power-broadened EPR line. The power broadening (i.e., multiplication by $1 - s$) leads to qualitative differences. For example, while the peak of the blue line in the
second row of the figure increased linearly with $\omega_1$, it now decreases as $1/\omega_1$. In the case of the orange line, the locations of its extrema are now shifted towards larger offsets ($\Omega \approx \pm\omega_1(T_{1S}/T_{2S})^{1/2}$) and their magnitude is approximately independent of $B_1$ ($\approx 0.5(T_{2S}/T_{1S})^{1/2}$, which equals $1/6 \approx 0.17$ for the choice of relaxation times in fig. 5). Clearly, the tail of the power-broadened dispersive (orange) component extends further into the range of interest for the solid effect at high mw frequencies than the tail of the absorptive (blue) component. One could thus expect that the path through $s_x^{\text{ss}}$ in fig. 3a (orange arrows)

contributes to $v_-$ more than the path through $s_y^{\mathrm{ss}}$ (blue arrows), simply because $s_y^{\mathrm{ss}}$ does not survive at offsets equal to the nuclear Larmor frequencies at high fields.

## 5.2 Generalized Bloch equations

The transfer functions indicated with colored arrows in fig. 4b depend on the auxiliary functions $F_{x,y}$ and $F_z$ (eq. (48)). These three complex-valued functions are plotted in the $\omega_I$-$\Omega$ plane in fig. 6a. In the plots, the angular frequencies are reported in
units of $R_{2S}$. Cross-sections at $\omega_I = 0, 0.5, 1.5, 3$ are drawn over the surfaces with solid black lines. The black lines at $\omega_I = 0$ show that the imaginary parts of $F_y$, $F_x$ and $F_z$ vanish, and their real parts become equal to $f_y$, $f_x$ and $f_z$ of the classical Bloch equations (cf. fig. 5, first two rows). In particular, at $\omega_I = 0$, $F_y$ and $F_x$ as functions of $\Omega$ are like the absorptive and dispersive components of the EPR line. When plotting $F_z$ we used $\omega_1 = 1.5$ (in units of $R_{2S}$). Because both the real and imaginary parts of $F_z$ decay very rapidly with increasing $\omega_I$, we also show the logarithm of the real part and the product of the imaginary part with $\omega_I$. These transformations make visible the small values of $F_z$ at large $\omega_I$.

In fig. 6b we show these functions against $\Omega$ at four different nuclear Larmor frequencies and, in the case of $F_z$, three different mw powers. In each case, the locations of the Larmor frequencies along the horizontal axis are indicated with vertical dashed lines. In the first and second rows we see $F_y$ and $F_x$, which do not change with mw power. The real and imaginary parts of $F_y$ (first row) look like the real and imaginary parts of two complex-valued Lorentzians centered at $\Omega = -\omega_I$ and $\Omega = +\omega_I$.
Indeed, with

$$L_\pm = [R_{2S} + \mathrm{i}(\omega_I \pm \Omega)]^{-1}, \tag{50}$$

it is straightforward to show that $F_y = (L_- + L_+)/2$. These Lorentzians have the same width as $f_y$ and $f_x$ of the classical Bloch equations (fig. 5, second row). The function $F_x$ in the second row of fig. 6b also has Lorentzian-like features centered at $\Omega = \pm\omega_I$, but the Lorentzian on the right is flipped around the horizontal axis. Indeed, it can be shown that $F_x = (L_- - L_+)/2$.
Differently from $F_{x,y}$, $F_z$ depends on $\omega_1$ (eq. (40)). In the last three rows of fig. 6b we plot $F_z(\Omega)$ for three different values of $B_1$, starting with $B_1 = 6\,\mathrm{G}$ (third row) and going down to $B_1 = 1.5\,\mathrm{G}$ (last row). The first thing to notice is that both the real (blue) and imaginary (orange) parts of this function decrease rapidly with increasing $\omega_I$, i.e., moving to the right in a given row. (The former as $1/\omega_I^2$ and the latter as $1/\omega_I$.) As all transfer functions (48) are proportional to $F_z$, we expect these to also decrease rapidly with increasing nuclear Larmor frequency.
At the lower mw powers and higher magnetic fields $F_z$ is seen to be dominated by its imaginary part, as its real part remains close to zero. At higher mw powers and lower magnetic fields ($B_1 = 6\,\mathrm{G}$, X and K bands, and $B_1 = 3\,\mathrm{G}$, X band) the real and imaginary parts are seen to be comparable in magnitude. Moving from the former to the latter regime, there is a major qualitative change: the features at $\Omega = \pm\omega_I$ shift towards the origin ($B_1 = 6\,\mathrm{G}$, K band, and $B_1 = 3\,\mathrm{G}$, X band) until they coalesce into a single line ($B_1 = 6\,\mathrm{G}$, X band).
In Paper II we calculate $F_z$ approximately using perturbation theory, and find

$$F_z \approx \frac{\cos^2 \alpha}{\tilde{R}_1 + \mathrm{i}\omega_I} + \frac{\frac{1}{2}\sin^2 \alpha}{\tilde{R}_2 + \mathrm{i}(\omega_I - \omega_{\mathrm{eff}})} + \frac{\frac{1}{2}\sin^2 \alpha}{\tilde{R}_2 + \mathrm{i}(\omega_I + \omega_{\mathrm{eff}})}, \tag{51}$$

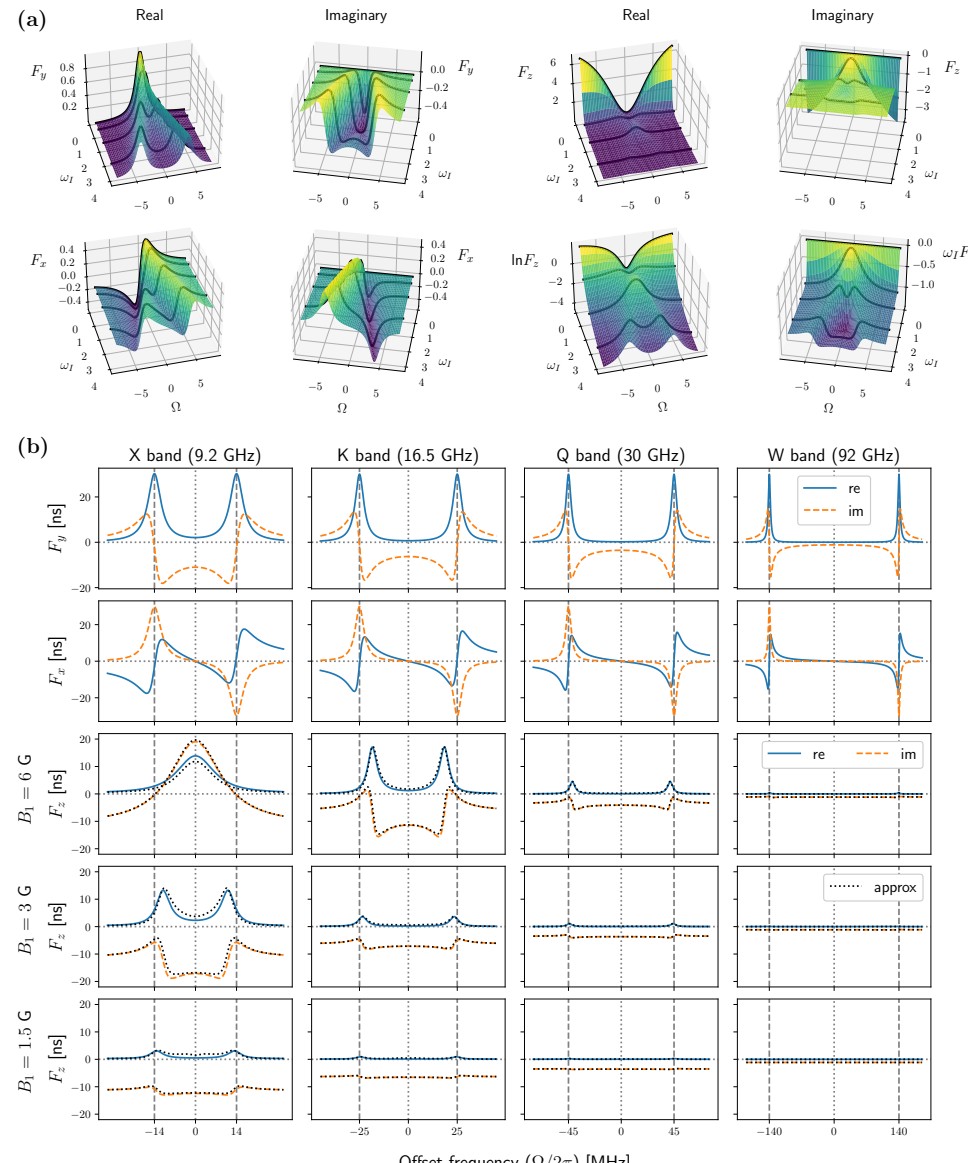

**Figure 6.** The functions $F_y$, $F_x$ and $F_z$ characterizing the steady state of the second set of Bloch equations. (**a**) Angular frequencies are measured in units of $R_{2S}$. $R_{1S} = R_{2S}/9$ as in the other figures. To calculate $F_z$ we used $\omega_1 = 1.5$, which for $T_{2S} = 60$ ns corresponds to $B_1 \approx 1.5$ G. Solid black lines are cross-sections at $\omega_I = 0, 0.5, 1.5, 3$. (**b**) Numerical parameters as in fig. 5. Recall that $B_1 = 6$ G corresponds to $\omega_1/2\pi = 16.8$ MHz.

where the frequency $\omega_{\text{eff}} = (\Omega^2 + \omega_1^2)^{1/2}$ corresponds to the effective magnetic field, $\alpha$ is the angle between this field and $B_0$, such that $\cos\alpha = \Omega/\omega_{\text{eff}}$ and $\sin\alpha = \omega_1/\omega_{\text{eff}}$, and

$$\tilde{R}_1 = R_{1S}(\cos\alpha)^2 + R_{2S}(\sin\alpha)^2$$

$$\tilde{R}_2 = R_{2S}[1 - (\sin\alpha)^2/2] + R_{1S}(\sin\alpha)^2/2. \tag{52}$$

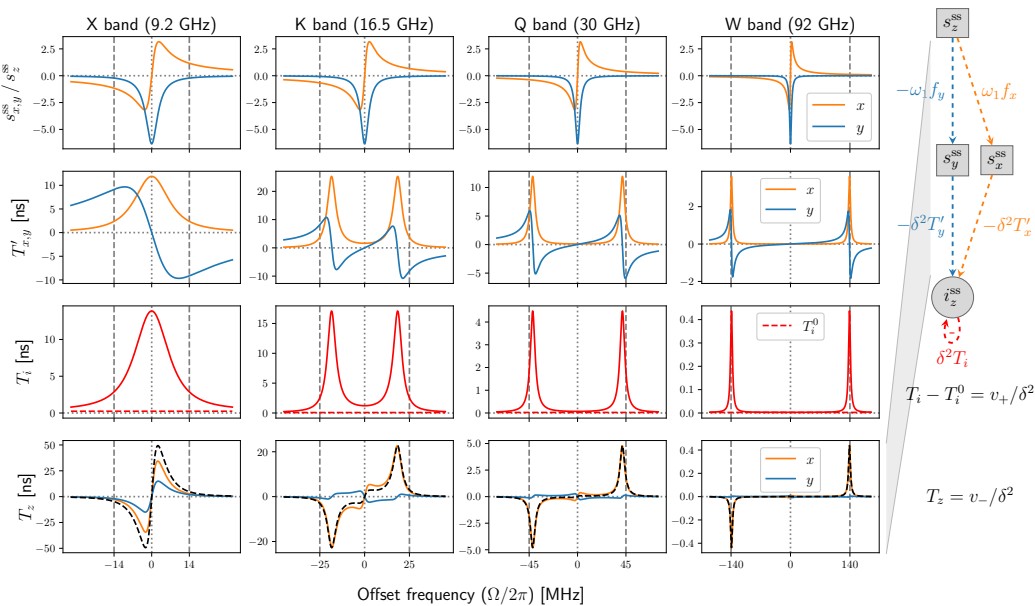

**Figure 7.** Transfer functions characterizing the steady state of the two coupled sets of Bloch equations. Used parameters: $B_1 = 6\,\mathrm{G}$, $T_{2S} = 60\,\mathrm{ns}$ and $T_{1S} = 9T_{2S}$.

This result is exact for $R_{1S} = R_{2S}$ and is perturbative in the difference of the two electronic relaxation rates (Sezer, 2023).

The approximation (51) is shown with dotted black lines in the last three rows of fig. 6b. It is seen to correctly capture both the shift of the peaks towards smaller offsets and their coalescence at $\Omega = 0$. Inspecting (51) we see that the dependence of $F_z$ on $\Omega$ comes from the second and third summands. The second summand is a complex-valued Lorentzian centered at $\omega_{\mathrm{eff}} = \omega_I$, which corresponds to the offsets $\Omega = \pm(\omega_I^2 - \omega_1^2)^{1/2}$. This explains the deviation of the maxima from the canonical solid-effect positions $\Omega = \pm\omega_I$ for $\omega_1 \approx \omega_I$. At X band, when $B_1 = 6\,\mathrm{G}$, $\omega_1$ is larger than $\omega_I$ and the two Lorentzians fuse together. It is noteworthy that the equality $\omega_{\mathrm{eff}} = \omega_I$, implied by the approximation (51), also arises as the matching condition of the pulsed DNP method known as NOVEL (nuclear orientation via electron spin locking) (Henstra et al., 1988; Henstra and Wenckebach, 2008; Jain et al., 2017).

We now turn to the transfer functions (48), which were depicted with colored arrows in fig. 4b. These are plotted in the second and third rows of fig. 7. As $T_i$ (solid red lines) is just the real part of $F_z$, it exhibits all the features that we already talked about when discussing fig. 6b. The dashed red lines in the third row of fig. 7 correspond to the mw-independent part of $T_i$, namely $T_i^0 = T_i(\omega_1 = 0)$, which contributes to the nuclear relaxation rate $w_1$ rather than to $v_+$ (eq. (47)). At the high mw field that we have used ($B_1 = 6\,\mathrm{G}$), $T_i^0$ is negligible compared to $T_i$ (solid red line), thus subtracting the relaxation would not make much of a difference. However, at lower mw powers the contribution of $T_i$ to thermal relaxation becomes comparable to the rest, and the correction makes a difference. (This can be seen in the bottom plot of fig. A1 where $B_1 = 1\,\mathrm{G}$.)

In the second row of fig. 7, the functions $T'_{x,y}$ resulted from the product of $F_{y,x}$ and $F_z$ (eq. (48)). Interestingly, their Lorentzian-like features are at the same frequency offsets as those of $T_i$, the real part of $F_z$. We observe that $T'_x$ (orange) and $T'_y$ (blue) are similar in magnitude. Thus, if $s_x^{ss}$ and $s_y^{ss}$ were comparable in magnitude, the contributions of the two parallel branches from $s_z^{ss}$ to $i_z^{ss}$ would be similar (see flow diagram in the right margin of fig. 7). We know, however, that $s_y^{ss}$ is much smaller than $s_x^{ss}$ at large offsets (fig. 7, first row), and so the path via $T'_x$ (orange) will contribute more.

Multiplying the functions $T'_{x,y}$ (fig. 7, second row) by the functions in the first row, we obtain the orange and blue lines in the last row of the figure. (The functions in the first row were shown before in fig. 5. They are plotted here again only for $B_1 = 6$ G. The four plots are identical to each other but appear different due to the different scales of the horizontal axes.) Comparing the first and second rows of fig. 7, we see that an odd/even function in the first row is multiplied by an even/odd function in the second row to produce the corresponding orange and blue lines in the bottom row. As a result, the contribution of both parallel paths from $s_z^{ss}$ to $i_z^{ss}$ (via either $s_x^{ss}$ or $s_y^{ss}$) is odd in $\Omega$. The cumulative transfer function of the two parallel paths (eq. (49)) is also plotted in the last row of fig. 7 with black dashed lines. At Q and W bands it is seen to be essentially identical to its first additive contribution $\omega_1 f_x T_x$ (orange line), which means that the electronic polarization is transferred to the nucleus almost entirely through the dispersive component $s_x^{ss}$.

In the light of this observation, we will now rewrite the cumulative transfer function $T_z$ (eq. (49)) as if the polarization was transferred only through the dispersive component. We start by observing that

$$F_y f_x + F_x f_y = F_y \frac{2R_{2S} + i\omega_I}{R_{2S} + i\omega_I} f_x = F'_y f_x, \tag{53}$$

where the last equality defines $F'_y$. The second $R_{2S}$ in the numerator of (53) comes from $F_x f_y$ and can be viewed as a "correction" to $F_y f_x$ due to $F_x f_y$. Introducing

$$T_x = \text{Re}\{iF_z(\omega_1 F'_y)\}, \tag{54}$$

(compare this $T_x$ with $T'_x$ in (48)), we rewrite (42) in a way that contains $s_x^{ss}$ but does not contain $s_y^{ss}$ as follows:

$$\dot{i}_z|_{coh}^{ss} = -(\delta^2 T_i)i_z^{ss} - (\delta^2 T_x)s_x^{ss}. \tag{55}$$

Note that this expression is exact, and does not result from simply dropping the last term in (42), which is proportional to $s_y^{ss}$, as the contribution of the path through $s_y^{ss}$ is taken into account in the definition of $T_x$.

Equation (55) is depicted in fig. 4d, which shows only one path from $s_z^{ss}$ to $i_z^{ss}$ going through $s_x^{ss}$. From fig. 4d,

$$v_- = (\omega_1 f_x)(\delta^2 T_x). \tag{56}$$

This factorization is revisited in Sec. 7.1.

## 6 Closer look at the rate constants

### 6.1 Relation to the classical rates

Here we show that the classical expression of the ZQ and DQ transition rates (eq. (28)) follows from the exact rates (eqs. (47) and (56)) when $\omega_1 \ll \omega_I$. To simplify the analysis, we take from the start a long electronic $T_1$ relaxation time, such that $R_{1S} \ll \omega_I$. This should be the case for high-field DNP in solids, where the electronic $T_1$ is at least a microsecond. In this case the function $F_z$ (eq. (40)) simplifies to

$$F_z \approx \frac{1}{\mathrm{i}\omega_I + \omega_1^2 F_y} = \frac{1}{\mathrm{i}\omega_I}\left(1 + \frac{\omega_1^2}{\mathrm{i}\omega_I}F_y\right)^{-1}. \tag{57}$$

For $\omega_1 \ll \omega_I$, to first order in $\omega_1^2$,

$$F_z \approx \frac{1}{\mathrm{i}\omega_I} + \frac{\omega_1^2}{\omega_I^2}F_y. \tag{58}$$

Note that, because the relaxation rate $R_{1S}$ was neglected, $T_i^0 = \mathrm{Re}\{F_z(\omega_1 = 0)\} = 0$. In other words, the contribution of the short path in fig. 3b (blue and red arrow) to the nuclear relaxation rate vanishes. From (47) and (56), retaining only terms of up to first order in $\omega_1^2$,

$$v_+ \approx \delta^2 \frac{\omega_1^2}{\omega_I^2}\mathrm{Re}\{F_y\}, \quad v_- \approx \delta^2 \frac{\omega_1^2}{\omega_I^2}\omega_I f_x \mathrm{Re}\{F_y \frac{2R_{2S} + \mathrm{i}\omega_I}{R_{2S} + \mathrm{i}\omega_I}\}. \tag{59}$$

To establish the equivalence of these expressions with (28), we need to show that $\mathrm{Re}\{F_y\}$ and $\omega_I f_x \mathrm{Re}\{F_y'\}$ equal, respectively, the sum and difference of two real-valued Lorentzians centered at $\Omega = \pm\omega_I$. For the complex-valued Lorentzians (50), we already observed that $L_- + L_+ = 2F_y$. One can also confirm that $\mathrm{Re}\{L_- - L_+\} = 2\omega_I f_x \mathrm{Re}\{F_y'\}$. Hence,

$$v_\pm \approx \delta^2 \frac{\omega_1^2}{\omega_I^2}\frac{1}{2}(\mathrm{Re}\{L_-\} \pm \mathrm{Re}\{L_+\}), \tag{60}$$

and thus

$$v_{0,2} \approx \frac{1}{8}(A_1^* A_1)\frac{\omega_1^2}{\omega_I^2}\mathrm{Re}\{L_\pm\}, \tag{61}$$

which is the classical result (28).

The sum and difference of the classical rates $v_2$ and $v_0$ is compared with the exact $v_\pm$ in the first two rows of fig. 8. Naturally, the Lorentzians associated with the classical rates remain centered at $\pm\omega_I$ even when the maxima of the exact rates 605 shift closer to each other at Q and K bands, and converge at X band. At high fields (e.g. W band), where $\omega_I \gg \omega_1$, the classical approximations work perfectly.

In the last row of fig. 8 we show the DQ-transition rate $v_2$. While, classically, it is always non-negative (black dashed lines), the exact rate deduced from $v_\pm$ (solid brown lines) is seen to become negative at some offsets. From the perspective of the rate-equation formalism, such negative rates are meaningless. In that sense, the description of the forbidden transitions in terms 610 of $v_\pm$ is more fundamental than their description in terms of $v_0$ and $v_2$.

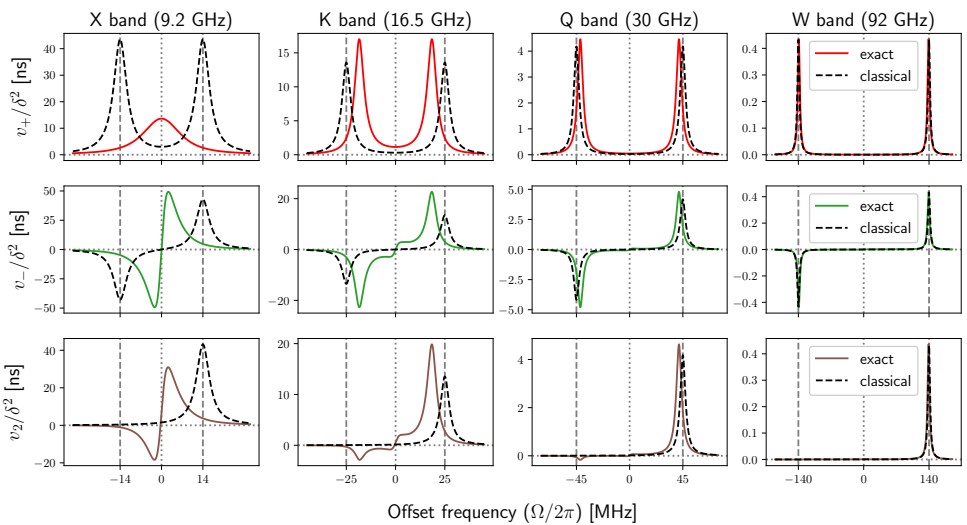

**Figure 8.** Forbidden-transition rates calculated either exactly (solid lines) or using the classical expression (28) with $v_\pm = v_2 \pm v_0$ (dashed lines). As in the previous figures, $B_1 = 6\,\text{G}$, $T_{2S} = 60\,\text{ns}$ and $T_{1S} = 9\,T_{2S}$.

## 6.2 Solid-effect DNP enhancement

The DNP enhancement of the solid effect (eq. (26)) can be written as the product of $|\gamma_S|/\gamma_I$ with the following two dimensionless factors:

$$p_X = \frac{R_{1I}/\delta^2}{R_{1I}/\delta^2 + (T_i - T_i^0)}, \qquad \frac{p v_-}{R_{1I}} = \frac{p T_z}{R_{1I}/\delta^2}, \tag{62}$$

which we have rewritten here in terms of the transfer functions $T_i$, $T_i^0$ and $T_z$. These transfer functions already appeared in the last two rows of fig. 7. Thus, to calculate the DNP enhancement, we only need to specify the ratio $R_{1I}/\delta^2$.

In the case of $\delta$, rather than calculating $A_1$ (eq. (30)) for some arbitrary inter-spin vector, let us average $A_1^* A_1$ over the entire 3D space. With $b$ denoting the so called "distance of closest approach" or "contact distance", and $N$ denoting the number of electron spins per unit volume, we have

$$\langle \delta^2 \rangle = \frac{1}{4} \langle A_1^* A_1 \rangle = D_{\text{dip}}^2 \frac{6\pi}{5} \frac{N}{3b^3}, \tag{63}$$

where, in this case, the angular brackets denote spatial averaging. We will use $b = 1\,\text{nm}$ and $N = 0.1\,\text{M}$ as representative, but otherwise arbitrary values.

While the average over 3D space in (63) is clear mathematically, it is important to understand that physically it implies fast spin diffusion (Wind et al., 1985). Since the nuclear polarization in solids is homogenized across the sample through spin diffusion, replacing the individual $\delta^2$'s of the nuclear spins by the average over all nuclei is only legitimate when spin diffusion is faster than the nuclear spin-lattice relaxation. In practice, spin diffusion is rather slow and is often the bottleneck for efficient

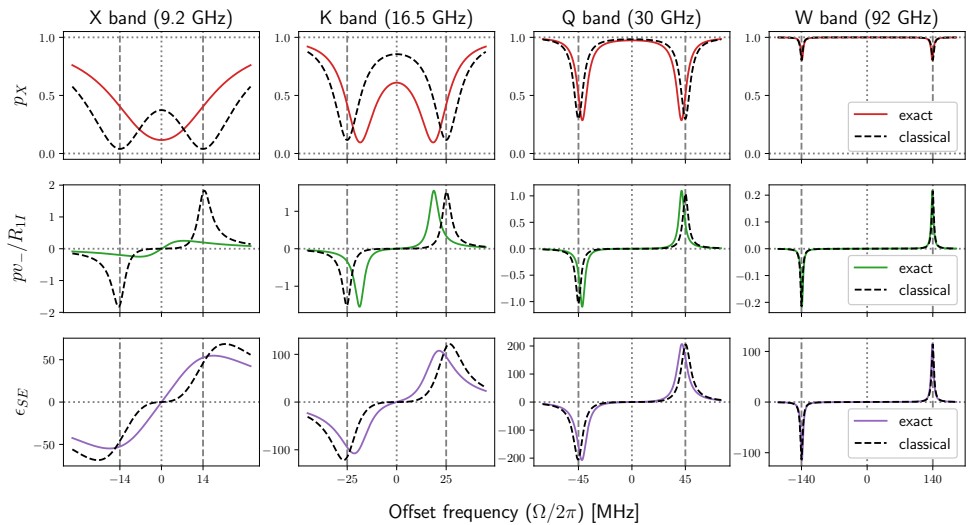

**Figure 9.** Decomposition of the DNP field profile ($\epsilon_{\mathrm{SE}}$) in terms of the multiplicative contributions $p_X$ and $pv_-/R_{1I}$. The new parameters used here are $T_{1I} = 30\,\mathrm{ms}$, $b = 1\,\mathrm{nm}$ and $N = 0.1\,\mathrm{M}$. Other parameters: $B_1 = 6\,\mathrm{G}$, $T_{2S} = 60\,\mathrm{ns}$ and $T_{1S} = 9\,T_{2S}$.

polarization transfer in solids (Hovav et al., 2011; Smith et al., 2012; Pinon, 2018). As a result, the DNP enhancement values that we will calculate with (63) are expected to be appreciably larger than what could be observed experimentally.

Similar considerations also apply for the choice of the nuclear spin-lattice relaxation time. In principle $T_{1I}$ will depend on the distance of the nucleus from the electronic spin, and thus will vary greatly across the sample. In the limit of fast spin diffusion, however, only its average value becomes relevant. In general, this time depends on the radical concentration and on the magnetic field $B_0$. However, for the purposes of illustration, here we take a generic numerical value of $T_{1I} = 30\,\mathrm{ms}$ across all mw bands. Again, this value is realistic but otherwise arbitrary.

Using $b = 1\,\mathrm{nm}$, $N = 0.1\,\mathrm{M}$ and $T_{1I} = 30\,\mathrm{ms}$ we find $R_{1I}/\langle\delta^2\rangle = 1.78\,\mathrm{ns}$. Let us visually compare this time scale with $(T_i - T_i^0) = v_+/\langle\delta^2\rangle$ by consulting the solid red line in the first row of fig. 8. We observe that at X and K bands the maxima of the red line are much larger than 2 ns, which means that the minima of $p_X$ will be close to zero. At Q band the maxima of the red line are comparable to 2 ns, and at W band they are much smaller. The minima of the nuclear cross-polarization factor are thus expected to be about one half and one, respectively. These expectations are confirmed by the maroon lines in the first row of fig. 9, which demonstrate that the ratio $p_X$ can substantially deviate from one at lower magnetic fields.

To estimate the expected magnitude of the second factor in (62), we need to compare the time scale $R_{1I}/\langle\delta^2\rangle = 1.78\,\mathrm{ns}$ with $pT_z$. While $T_z$ was shown with black dashed lines in the bottom row of fig. 7, now it has to be multiplied by the electronic polarization factor in the top row of fig. 5. From the line for $B_1 = 6\,\mathrm{G}$ in this row, we see that $T_z$ will be significantly suppressed at X band, so it is hard to judge how the reduced value will compare with 1.78 ns. At Q band, $T_z$ will be reduced by a little more than a factor of two, which will make its peak in fig. 7 comparable to $R_{1I}/\langle\delta^2\rangle$. At W band, where the factor $p$ is about 0.9, $T_z$

will be only slightly reduced, so its peak is expected to be about one fifth of 1.78 ns. Again, these estimates are confirmed by the green lines in the second row of fig. 9.

The last row of fig. 9 shows the product of the first two rows times $|\gamma_S|/\gamma_I$, assuming a proton spin. The result is the solid-effect DNP enhancement (eq. (26)). In the figure we have also shown the factors predicted by the classical expression of the rates (eq. (28)) with black dashed lines. While there are quantitative differences between the exact calculations and the classical

approximation, the magnitudes of the DNP enhancements in the two cases are, in fact, comparable. A closer look reveals that, for the specific $B_1$ and relaxation times used in the calculations, the classical description of the solid effect (eq. (28)) works perfectly at Q band and at larger mw frequencies. (In fig. A2 we show that by reducing the mw power to $B_1 = 1$ G the classical expressions are also perfect at X band.) The amplitudes of the maximum enhancements at the four mw bands are roughly in the ratios $1:2:4:2$ (X:K:Q:W). On the other hand, considering the inverse dependence on $\omega_I^2$, we expect the ratios $100:40:10:1$.

These expected ratios are indeed observed at the much lower mw power of $B_1 = 1$ G (fig. A2, lower plot). Comparison of figs. 9 and A2, shows that increasing $B_1$ increases the amplitudes of the maximum enhancements at W and Q bands, but reduces the enhancement at X band. Such reduction of the solid-effect DNP enhancement with increasing $B_1$ has been reported at X band (Neudert et al., 2016).

# 7   Concluding discussion

## 7.1   Refactorization of the polarization transfer

When $p_X \approx 1$ (eq. (26)), e.g., at high magnetic fields (fig. 9, W band) and lower mw powers (fig. A2, lower half), the DNP enhancement of the solid effect is

$$\epsilon_{\text{SE}} \approx (pv_-)T_{1I}|\gamma_S|/\gamma_I \qquad (p_X \approx 1). \tag{64}$$

Since $T_{1I}$ is easily accessible experimentally, $pv_-$ is the only non-trivial factor in (64). From fig. 4a we know that $p$ relates $s_z$

at steady state to $s_z^{\text{eq}}$, and from fig. 4c we know that $v_-$ relates the time derivative of $i_z$ at steady state to $s_z$. Hence, the product $pv_-$ relates the time derivative of $i_z$ directly to the electronic Boltzmann polarization $s_z^{\text{eq}}$, as shown graphically in the left half of fig. 10.

Since, by construction, the rate equations of the polarizations do not model the dynamics of the coherences, their steady state balances the rates of mw excitation only against the longitudinal (i.e., spin-lattice) relaxations. The polarization factor $p$

quantifies this balance for the allowed EPR transition (eq. (3)). Because the rate equations work only with the polarizations, all dynamical variables between $s_z$ and $i_z$ in fig. 2d are lumped into the rate constant $v_-$. Classically, this rate constant ($v_- = v_2 - v_0$) is obtained by calculating the rates of the ZQ ($v_0$) and DQ ($v_2$) transitions using first-order perturbation theory. From this point of view, decomposing the product $pv_-$ into the factors $p$ and $v_-$ is natural. The offset dependence of these two factors was visualized in fig. 5 (top row) and fig. 8 (middle row, black dashed lines). The curves for $B_1 = 6$ G and W band are

reproduced on the left-hand side of fig. 10 (black and green lines).

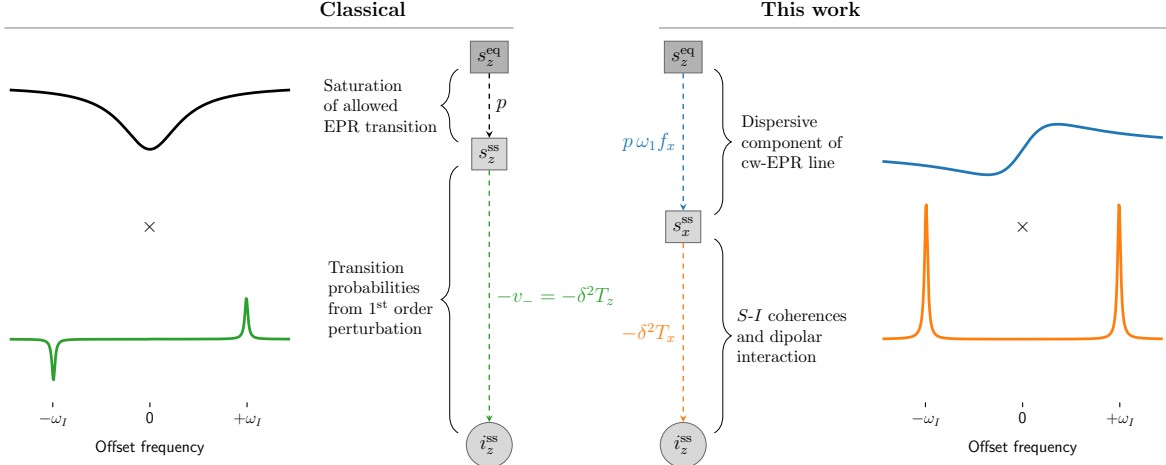

**Figure 10.** Two ways of decomposing the effect of the electronic Boltzmann polarization ($s_z^{\mathrm{eq}}$) on the steady-state nuclear polarization ($i_z^{\mathrm{ss}}$). The classical way (left) partitions this effect into the factors $p$ and $v_- = v_2 - v_0$, which reflect respectively the saturation of the allowed EPR transition and the excitation of the forbidden DQ ($v_2$) and ZQ ($v_0$) transitions. Alternatively (right), the same effect can be written as the product of the dispersive component of the power-broadened EPR line ($s_x^{\mathrm{ss}}/s_z^{\mathrm{eq}}$) and the rate constant $\delta^2 T_x$. The latter characterizes the steady state of the electron-nucleus coherences without any contribution from the purely electronic coherences.

In contrast to this classical approach, here we considered the complete spin dynamics of relevance to the solid effect, including the dynamics of the coherences (fig. 2d). The analysis was simplified by the realistic assumption that the electronic dynamics was not affected by the dipolar interaction with the nuclear spins. Thus, in our description, the purely electronic degrees of freedom constitute an isolated dynamical system, which influences the other dynamical variables but is not affected
by them.

This division of the complete dynamical system into a purely electronic part and the rest calls for a similar separation of the product $pv_-$ in (64) into an electronic part and a mixed electron-nucleus part. Such factorization of $pv_-$ is illustrated in the right half of fig. 10, where the purely electronic part is identified with the dispersive component of the EPR line. This would be the out-of-phase cw-EPR spectrum recorded under the same mw power as used in the DNP experiment. Then, from (56),
the second factor is recognized to be $\delta^2 T_x$, where $\delta^2$ accounts for the strength of the dipolar interaction (eq. (41)) and $T_x$ takes care of the interconnections between the relevant electron-nucleus coherences at steady state (eq. (54)). The offset dependence of the dispersive EPR line was visualized before in fig. 5 (bottom row). The curve for $B_1 = 6\,\mathrm{G}$ is reproduced on the right-hand side of fig. 10 (blue line). The curve below it (orange line) corresponds to $\delta^2 T_x$ at W band, which is essentially the same as $\delta^2 T_x'$ that was shown in the second row of fig. 7 since at this high magnetic field $T_y'$ contributes negligibly little.
Because, as already illustrated above (fig. 9, middle row, W band), the classical approach and our new approach lead to the same product $pv_-$, the new factorization on the right-hand side of fig. 10 may appear as a purely mathematical exercise of little practical interest. Note, however, that recognizing the dispersive EPR line as contributing multiplicatively to the DNP

enhancement suggests that the dispersive extrema could become visible in the field profile of $\epsilon_{SE}$, provided that they are not fully suppressed by the factor $\delta^2 T_x$. Such possibility is completely missing in the classical description on the left-hand side of fig. 10, where any reference to the dispersive EPR line and its extrema is irrelevant.

In Paper II we show that, in liquids, the random modulation of the dipolar interaction broadens the lines of the factor $\delta^2 T_x$ (fig. 10, orange line). When the tails of these broadened lines reach the extrema of the dispersive EPR line (blue line), the enhancement field profile exhibits features that are reminiscent of the DNP effect known as thermal mixing (Kuzhelev et al., 2022). These features are a direct manifestation of the dispersive EPR line in the DNP spectrum (Sezer, 2023).

## 7.2 On the birth of the solid effect

The issue of *Comptes rendus* from April 9, 1958, contained the article "Effect of nuclear polarization in liquids and gases adsorbed on charcoal" by Erb, Motchane and Uebersfeld (Erb et al., 1958a). It reported enhancements of the proton NMR signal of benzene upon mw irradiation of the EPR line of charcoal. The enhancements were positive at fields larger than the EPR resonance position and negative at smaller fields. Because fields symmetrically displaced from the resonance yielded the same magnification factor, the enhancement profile was odd in the field offset and resembled the dispersive component of the EPR line. The similarity between the two prompted the authors to augment the Solomon equation (Solomon, 1955) with two new terms proportional to $s_x$ and $s_y$ (Erb et al., 1958a):

$$\dot{i}_z = \lambda(i_z - i_z^{eq}) + \mu(s_z - s_z^{eq}) + \nu s_x + \rho s_y. \tag{65}$$

Taking into account that "under saturation conditions $s_y = 0$" the authors arrived at

$$\dot{i}_z = \lambda(i_z - i_z^{eq}) + \mu(s_z - s_z^{eq}) + \nu s_x. \tag{66}$$

Assuming $\mu$ was small in their case, they solved (66) at steady state as

$$i_z^{ss} = i_z^{eq} - (\nu/\lambda)s_x^{ss}, \tag{67}$$

which explained the similarity between the field profile of the enhancement and the dispersive EPR line.

Intriguingly, with $\mu = 0$, the phenomenological equation (65) is mathematically identical to (42), which expressed the time derivative of $i_z$ at steady state as a linear combination of $i_z$, $s_x$ and $s_y$. The argument of Erb et al. (1958a) that the contribution of $s_y$ could be neglected, which let to (66), is justified by our analysis. Specifically, in the last row of fig. 7 we observed that the contribution of the absorptive component $s_y$ to the rate constant $v_-$ was smaller than that of the dispersive component $s_x$. Moreover, we showed that the mathematically identical equation (55) was, in fact, exact within the framework of our treatment. Thus, the phenomenological equation (66) produces the correct steady state when its coefficients are selected as $\nu = -\delta^2 T_x$ and $\lambda = -\delta^2(T_i - T_i^0)$.

The next installment of *Comptes rendus* from April 14, 1958, contained Abragam and Proctor's report "A new method for dynamic polarization of atomic nuclei in solids" (Abragam and Proctor, 1958), which was printed 132 pages after Erb et al. (1958a). This seminal contribution provided the modern theoretical understanding, and subsequently also the name, of the

solid-state effect of dynamic nuclear polarization (DNP). In particular, the authors argued that the excitation of the forbidden
transitions $(++) \rightleftharpoons (--)$ and $(+-) \rightleftharpoons (-+)$, which become weakly allowed because the dipolar coupling yields mixed states
of the form $(--) + q(-+)$, could be used for DNP. ($\pm$ are the states of the two spin types, both taken as 1/2 for simplicity.)
As an experimental verification of the theoretical proposal, the Boltzmann polarization of $^{19}$F nuclei was used to enhance the
NMR signal of $^6$Li in a LiF monocrystal, thus demonstrating polarization transfer from nuclei with larger to nuclei with smaller
gyromagnetic ratios (i.e., a *nuclear* solid effect).

One month and a half after Abragam and Proctor's report, in the May 28, 1958 issue of *Comptes rendus*, Erb, Motchane
and Uebersfeld published another report with the lengthy title "On a new method of nuclear polarization in fluids adsorbed on
charcoal. Extension to solids and in particular to irradiated organic substances" (Erb et al., 1958b). There, the authors state (our
translation)

> The experiments [Erb et al. (1958a)] had been carried out with charcoal whose half-linewidth was 5 gauss and the
> multiplication factor seemed to reproduce the paramagnetic dispersion curve.

> The new experiments [...] indicated that the increase in polarization of the proton in the adsorbed fluid is max-
> imum in all cases, when the electronic and nuclear frequencies are chosen such that the nuclear resonance field
> differs from the electron resonance field $\delta H = \pm 5$ gauss (within 10%).

> These results support the suggestion of Abragam that the new theory of Abragam and Proctor on the nuclear
> polarization in solids (Abragam and Proctor, 1958) must apply to these new phenomena, and invalidates the inter-
> pretation proposed previously (Erb et al., 1958a).

> The value of 5 gauss found in the case of the proton indeed corresponds to the value deduced from the theoretical
> formula $H_0 \pm \delta H = (\omega \pm \omega_N)/\gamma_e, \ldots$.

This seems to have sealed the fate of the insightful observation of Erb et al. (1958a) that the odd parity of the solid-effect DNP
field profile resembles the dispersive component of the EPR line.

With the understanding developed in the 65 years since these first publications on the solid effect, the additional transverse
terms in (65) appear strange, and even disturbing. Nevertheless, our analysis showed that in one specific regime—steady
state—equation (65) is exact. Admittedly, because of the algebraic relationships between all dynamical variables at steady
state, the transverse components in (65) can be expressed in terms of the longitudinal component, as we did when going
from (42) to (43). Such mathematical manipulation, however, only highlights the fact that the value of any description of spin
dynamics by rate equations, independently of whether it contains transverse components or not, lies in the proper selection of
the phenomenological rate constants. In this paper, we departed from the classical approach of identifying these rate constants
with the transition probabilities per unit time. Instead, completely disregarding the dynamical aspect of the rate equations, we
selected the phenomenological rate constants by requiring that the steady state of the exact quantum dynamics is correctly
reproduced.

By writing the rate equation of the nuclear polarization with explicit dispersive component (eq. (66)), Erb et al. (1958a)
reached the conclusion that the DNP enhancement depends *multiplicatively* on $s_x$ (eq. (67)). This conclusion is confirmed by

our analysis. Indeed, from the new perspective illustrated on the right-hand side of fig. 10, the DNP field profile acquires its odd parity in $\Omega$ directly from the dispersive component of the EPR line (blue line), exactly as intuited by Erb et al. (1958a). Certainly, one could explain the odd parity of the solid-effect DNP enhancement in various other ways that do not involve the dispersive EPR line, as has been done in the past 65 years. The validity of these other explanations, however, does not invalidate the intuition of Erb, Motchane, and Uebersfeld.

## 7.3 Conclusion

In this paper we developed a novel way of thinking about the solid effect, which was grounded in the dynamics of the spins at steady state. The main insight of our dynamical description relates to the role of the coherences.

While our analysis focused on the solid effect and the Hamiltonian (29), the systematic procedure for deriving the relevant equations of motion under a given spin Hamiltonian (Sec. 3.2), and the developed graphical representations to visualize the interplay of these equations (Sec. 4) and their steady state (Sec. 5), should be applicable to other related effects with different Hamiltonians.

The classical explanation of the solid effect in terms of state mixing (Abragam and Proctor, 1958) is static in nature and is thus hard to generalize to liquids where the dipolar interaction fluctuates randomly due to molecular motions. The time-dependent description of the solid effect developed here naturally accommodates such stochastic modulation of the parameters of the Hamiltonian, in a way similar to the treatment of relaxation in liquids (Abragam, 1961, Ch. VIII). In the companion paper (Sezer, 2023), the formalism is extended to the solid effect in liquids, and its predictions are validated against recent DNP experiments at J band (Kuzhelev et al., 2022).

## Appendix A: Additional figures

The numerical examples in the paper were for the excessively high mw field of $B_1 = 6\,\text{G}$, which is reachable with a custom-designed resonance structure (Denysenkov et al., 2022). As the modern-day DNP experiments in solids are generally performed without a mw resonator, here we show numerical examples for the lower fields of $B_1 = 3\,\text{G}$ and $B_1 = 1\,\text{G}$. Although these are still likely an order of magnitude larger than what is used in practice, the figures aim to illustrate how some of the features discussed in the paper progressively change upon reduction of $B_1$.

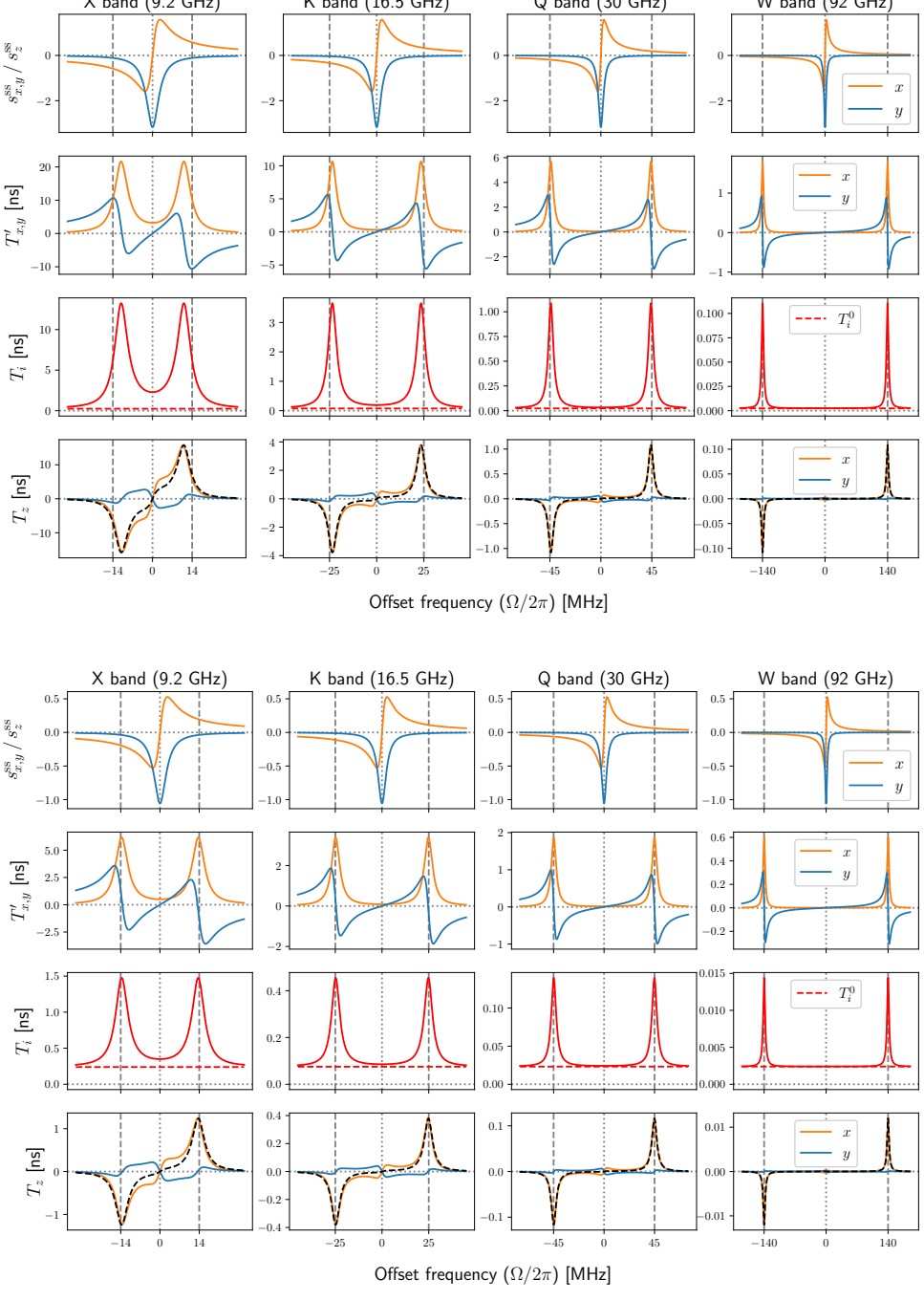

**Figure A1.** Same as fig. 7 with smaller mw fields of $B_1 = 3\,\text{G}$ (top) and $B_1 = 1\,\text{G}$ (bottom).

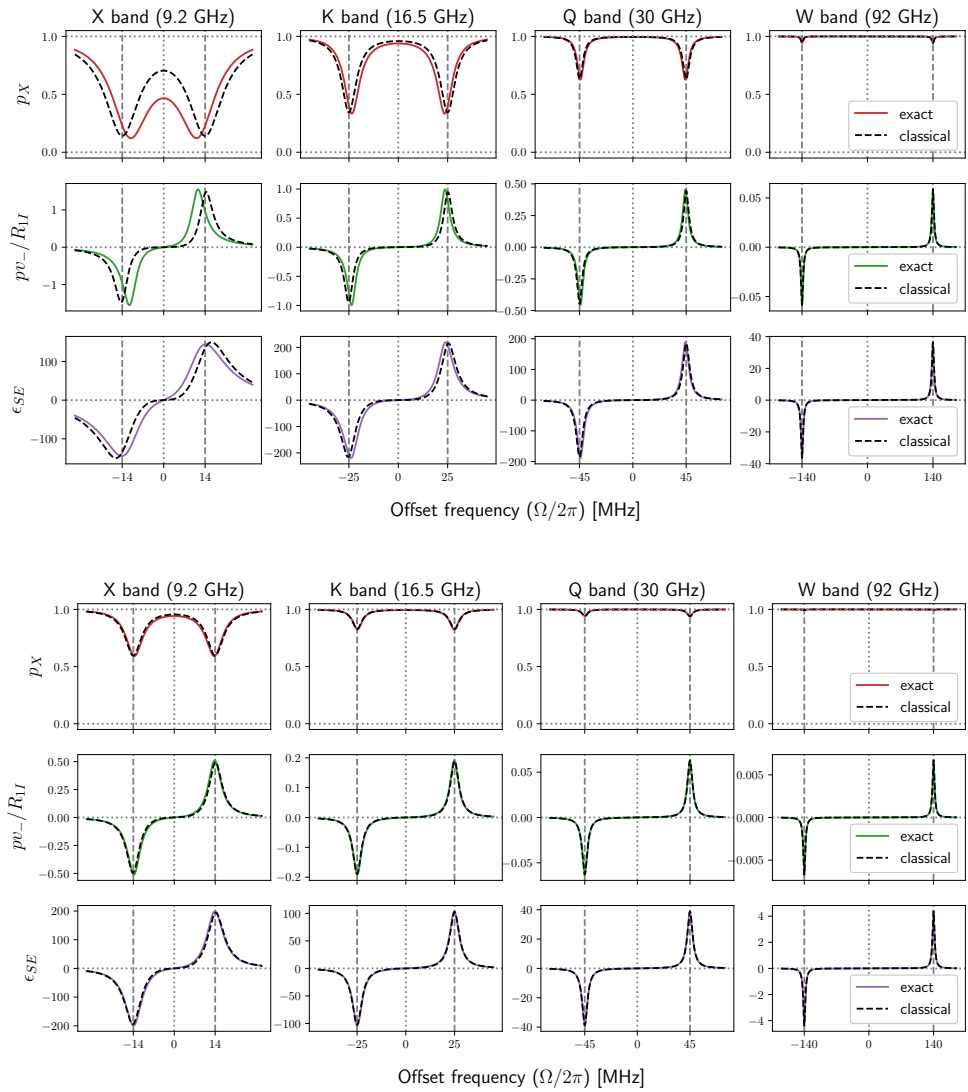

**Figure A2.** Same as fig. 9 with smaller mw fields of $B_1 = 3\,\text{G}$ (top) and $B_1 = 1\,\text{G}$ (bottom).

*Author contributions.* DS developed the theory and wrote the paper.

*Competing interests.* No competing interests are present.

*Acknowledgements.* Thomas Prisner is gratefully acknowledged for providing the scientific environment that enabled the reported research. Numerous discussions with Thomas Prisner, Andriy Marko, Vasyl Denysenkov, Andrei Kuzhelev, and Danhua Dai were instrumental in developing the presented ideas. Funding was provided by the German Research Foundation (DFG grant 405972957).

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
