# Peer review of "Dynamic view of the solid-state DNP effect"

_Magnetic Resonance, 2023_

## Author Comment (AC1)

**Response to the reviewers of the the manuscript**
**Dynamic view of the solid-state DNP effect**

Deniz Sezer

April 24, 2023

I would like to thank the reviewers for critically reading the manuscript and for providing valuable feedback on how to improve the presentation.

The main criticism of all reviewers is that the organization of the paper was confusing, especially because the reader was expected to switch back and forth between dubious interpretation of historical papers, manipulation of equations, and diagrams intended to visually represent these equations. Because none of the three referees had specific questions concerning the scientific content of the paper, below I first reproduce all comments, and then outline the ways I restructured the manuscript in line with the extensive constructive feedback. When reproducing the comments, I have taken the liberty to highlight some parts by coloring the identified strengths of the manuscript in blue, the identified weaknesses in magenta, and the remarks related to the historical perspective in cyan.

**1 Anonymous**

The reviewed manuscript describes an analytically derived description of the so-called solid-state DNP effect. As stated by the author, his main goal was to rehabilitate the approach proposed in the 1958 paper by Erb et al. in Comptes rendus "The nuclear polarization effect in liquids and gases adsorbed on charcoal," in which the antiphase structure of the DNP spectrum, later called the solid effect, is interpreted by the determining contribution of equilibrium dispersion of electron spin magnetization. I believe that this cannot be considered the purpose of the manuscript, since the dispersion extremums in their approach are determined solely by the transverse electron relaxation time $T_2$, whereas the proper mechanism of the solid effect gives a description of the two extremum positions with their dependence on the nuclear Zeeman frequency. Moreover, the approach of Erb et al. (1958), as they themselves wrote, was purely phenomenological, in contrast to the correct explanation given by Abraham and Proctor in the same issue of Comptes rendus (1958). Nevertheless, the result of the peer-reviewed paper may be considered to be that the author has shown that the antiphase $(+/-)$ character of the solid-state effect spectrum is determined by the antiphase character of the magnetization $s_x$ of the stationary electron. Also I find it a very interesting result of the manuscript that the author has reduced the solid-state effect problem to a closed system of equations for the mean values of seven spin operators to obtain the stationary nuclear polarization. Among these seven mean values of the operators there are three projections of electron spin, for which stationary values can easily be found from the Bloch equations without taking into account the coupling of electron and nucleus spins. This is reasonable, because the equilibrium of the electron spin is reached much faster than that of the nuclear spin.

At the same time, I consider completely unjustified and unnecessary all the graphical representations of the differential equations, with which the work is simply overloaded. In my opinion, they

provide no new information and distract attention. Extremely much space is given to the usual Bloch equations for the projections of the electron spin, taking into account the microwave field and the calculation of their stationary values, as well as the determination of transition probabilities for the kinetic transition scheme, these well-known solutions are given in all textbooks on magnetic resonance. I believe that the article simply needs to be very significantly reduced as follows:

1. Bloch equations and resulting transition probabilities should not occupy more than half a page;

2. Exclude graphical diagrams of all differential equations duplicating all equations (i.e. not only Bloch equations), since they do not carry any additional information, unlike, for example, the well-known Feynman diagrams.

Thus, the paper can be published only after major revision.

**2    Anonymous**

In this manuscript, the author is deriving a comprehensive description of the solid effect DNP mechanism based on a steady-state description of, first, rate equations and, second, spin dynamics. It goes beyond many of the previously published theoretical works as the latter are often simplifying the underlying problem by neglecting excitation of single quantum electron spin coherence. The author's analytical description therefore allows a mechanistic explanation and understanding of broadening occurring under strong microwave fields at relatively low fields, where the EPR nutation frequency is comparable in magnitude to the nuclear Larmor frequency. This is where from my perspective the strength of the paper lies: it nicely shows and explains the deviation from the most often used perturbation approach, where microwave irradiation is assumed to be small compared to the nuclear Larmor frequency and the accompanied effects in DNP profiles and enhancement factors.

However, in the current form I found it in many parts frustrating to follow the paper as **the actual motivation is unclear and shifts over the course of the text**. Particularly in the beginning the most focus is laid on a statement of Erb at al. from the paper describing the first observation of solid effect on adsorbed liquids. First, the description of "DNP in liquids via solid effect" in the very first sentence of the abstract skews the expectation in a wrong direction (i.e. DNP in liquids with HR-NMR). Second, the focus set on arguing that Erb et al.'s statement is actually correct takes a large part of the reader's attention away from the actual scientific content of the work at hand. With our modern understanding of the solid effect and off-resonance effects, the similarity in the odd parity of the DNP field profile and the shape of the EPR dispersion line should be nothing surprising. One could provide many different but all similarly valid reasonings why both spectra have the same parity, starting from an interaction frame treatment to energy conservation in an isolated system of an electron and a nucleus (and the required photon). Therefore, I do not see how this topic as the central part of the paper provides the necessary scientific novelty for publication. In contrast, it should be made clearer in the introduction, in which regard other descriptions fail and how this work provides better insight into the solid effect dynamics.

Additionally, the graphical representation of the spin dynamics (while undoubtedly carrying some didactical value) may confuse the reader as it **requires a constant switching from equation based description to graphical representation**, back and forth throughout the theoretical derivations. Therefore, I propose to remove this aspect from the main text and place it either in the SI or publish it in a more educational context in a separate manuscript.

Finally, figures 11 and 12 are not shown close to the point of first mention and are not in the same order as they are referenced in the text (figure 11 is first referenced after figure 8). They could be moved to the SI.

As mentioned above, this manuscript provides a sound, interesting and comprehensive description of the solid effect under a broad range of experimental conditions (low to high power, low to high field) and therefore could be publishable in MR after major revisions regarding the points I have mentioned above.

**3    Malcolm Levitt**

This is a rather eccentric but interesting paper on the theoretical description of "solid effect" DNP. In common with the other referees I found my appreciation of the paper strongly hampered by its frustrating style. For example, the electronic analogy of "band pass filters" may mean something to the author and perhaps to others in the field but I found it just distracting and unhelpful, and did not aid my understanding of what the referee is trying to say, one bit. Similarly the discussion around eq.1 with its strange additional terms just seems like time wasted on very old speculations which proved to be a dead end. If the theory of the authors does lead to something equivalent to eq.1 then **that can be stated in the discussion at the end of the paper** as an interesting historical analogy but certainly not as a central focus to assist modern understanding. In summary I found the paper extremely hard to understand and was left constantly frustrated that the author did not present his arguments in a straightforward, linear, way without historical or diagrammatic distractions. The author failed to convince me that the complicated diagrams are of any substantial help - on the contrary. In summary I think this is potentially an interesting and significant piece of work but it badly needs to be shortened considerably, with **a linear straightforward presentation**, and with the confusing and unhelpful historical aspects brushed aside.

**4    Description of the implemented changes**

The manuscript underwent two major organizational alterations.

1. The historical discussion and the pointless analogy with electrical filters was removed entirely from the Introduction. A completely new Introduction was written, which hopefully spells out more clearly both the motivation and the contribution of the paper, thus addressing the justified concern of the second reviewer that "the actual motivation is unclear".

2. The derivations are now presented in a "straightforward, linear way" (Malcolm Levitt) in the first half of the paper (Secs. 2 and 3), without interruption by any figures and diagrams. This should also address the fair criticism of the second reviewer that the previous version required "a constant switching from equation based description to graphical representation".

The other organizational changes are as follows:

- As suggested by Malcolm Levitt, the analogy between the results of the paper and the phenomenological equations of Erb, Motchane and Uebersfeld [1] is now moved to the Discussion (Sec. 7.2).

- The diagrammatic representation of coupled differential equations is now collected in a single section (Sec. 4), which starts with a brief motivation and ends with a concrete result that would be difficult to reach without the insight from the visual inspection of the equations.

- Similarly, the analysis of the algebraic relations between the variables at steady state is collected in a separate section (Sec. 5).

- The very little that is left from the idea of "filters" is presented (hopefully, in a more intelligible way) as a summarizing overview towards the end (fig. 10), before the historical discussion and the conclusion.

Finally, I would like to address the two specific requests of the first reviewer.

1. Bloch equations and resulting transition probabilities should not occupy more than half a page;

   The Bloch equations are discussed in the paper because they constitute an integral part of the full dynamics that describes the solid effect. Although the derivations of the coherent part of these equations and of their steady state are textbook material, their repetition in the paper is intended to serve a pedagogical role, as exactly the same steps are then followed for the two-spin system.

   Nevertheless, I made two revisions:

   - Figure 4, which repeated the derivation of the steady state in graphical form, is scrapped. (This was, indeed, a very confusing figure.)
   - To better justify my derivation of the steady state of the Bloch equations, I now use exactly the same elimination of variables when deriving the steady state of the four-level system. In other words, I no longer use matrix notation. (Removing the matrix inversion simplified the presentation.) The parallel derivation in the two cases is expected to clarify exactly in what way the functions $f_x$, $f_y$ and $f_z$ that appear in the Bloch equations are generalized by the functions $F_x$, $F_y$ and $F_z$.

2. Exclude graphical diagrams of all differential equations duplicating all equations (i.e. not only Bloch equations), since they do not carry any additional information, unlike, for example, the well-known Feynman diagrams.

   I think that the comparison with Feynman diagrams is distracting.

   In my opinion, it is more helpful to compare the diagrams in the manuscript with the diagrammatic representation of chemical reactions. In fact, when I first drew the diagram of the Bloch equations, I though that I was using the standard representation of coupled chemical reactions. Only after drawing the diagram I realized that my arrows could also have negative weights and, more importantly, when translating the diagram back into differential equations, I realized that my arrows did not deplete the node from which they originated. Nevertheless, in spite of these differences, the visual representation that I propose is intended to serve exactly the same purpose as the representation of chemical reactions.

   In the case of two or three coupled chemical reactions, the diagrammatic representation likely does not help our understanding. However, to represent even a relatively simple metabolic pathway, like the glycolysis pathway with ten reactions, for example, one always draws the reactions as arrows. There is perhaps not a single biochemistry textbook which writes down the coupled differential equations that are implied by these reactions. In fact, the best way to comprehend how the kinetic variables are connected to each other by the various interactions is to draw these variables as nodes and the interactions as links. Such representation readily

reveals the connectivity of the network in a visual way, certainly much better than writing down all the coupled differential equations and looking at them.

In my experience, as soon as I moved beyond the familiar Bloch equations, the diagrammatic representation served as a map that allowed me to quickly get oriented in the new terrain. I could visually identify the coherences that get coupled by, for example, the dipolar interaction or the microwave excitation. On several occasions, the repeated patterns and symmetries in the diagram allowed me to identify wrong prefactors or wrong signs in my derivation of the equations.

The diagrammatic representation is also helpful for asking and thinking about higher-level questions. For example, one unsolved question in the field relates to the mechanism of the Overhauser effect in insulating solids. I have long looked at the diagrammatic representation of the seven coupled differential equations describing the solid effect, trying to think of a way of reconnecting the coherences such that the path from the electronic polarization to the nuclear polarization is no longer odd in the offset but even (as needed by the Overhauser effect). Without a diagram, it would be impossible to reason about such global properties related to the topology of the connections.

In summary, I am convinced that the graphical representation of the coupled differential equations that I have in the paper is an extremely useful tool, which deserves to be promoted and used more broadly in magnetic resonance, even if only for pedagogical reasons. Personally, I view this graphical representation as an even more important contribution of the paper than the specific derivation regarding the solid effect, since it has the potential to be useful to more people and in other contexts.

**References**

[1] E. Erb, J.-L. Motchane, and J. Uebersfeld, "Effet de polarisation nucléaire dans les liquides et les gaz adsorbés sur les charbons," *Compt. rend.*, vol. 246, pp. 2121–2123, 4 1958.